# Numerical study on the response of the largest lake in China to climate change

Dongsheng Su[1, 2], Xiuqing Hu[3], Lijuan Wen[1], Shihua Lyu[1,5], Xiaoqing Gao[1], Lin Zhao[1], Zhaoguo Li[1], Juan Du[1, 2], Georgiy Kirillin[4]

[1]Key Laboratory of Land Surface Process and Climate Change in Cold and Arid Regions, Northwest Institute of Eco-Environment and Resources, Chinese Academy of Sciences, 730000 Lanzhou, China
[2]University of Chinese Academy of Sciences, 100049 Beijing, China
[3]Key Laboratory of Radiometric Calibration and Validation for Environmental Satellites, National Satellite Meteorological Center, China Meteorological Administration, 100081 Beijing, China
[4]Department of Ecohydrology, Leibniz-Institute of Freshwater Ecology and Inland Fisheries (IGB), 12587 Berlin, Germany
[5]Plateau Atmosphere and Environment Key Laboratory of Sichuan Province, School of Atmospheric Sciences, Chengdu University of Information Technology, 610225 Chengdu, China

*Correspondence to*: Lijuan Wen (wlj@lzb.ac.cn)

**Abstract.** Lakes are sensitive indicators of climate change. There are thousands of lakes on the Tibetan Plateau (TP), more than 1200 of them having an area larger than 1 km$^2$, which respond quickly to climate change, but few observation data of lakes are available. Therefore, the thermal condition of the plateau lakes under the background of climate warming remain poorly understood. In this study, the China Meteorological Forcing Dataset developed by Institute of Tibetan Plateau Research, Chinese Academy of Sciences (ITPCAS), MODIS Land Surface Temperature (LST) data and buoy observation data were used to evaluate the performance of lake model FLake, extended by simple parameterizations of salinity effect, for brackish lake, and reveal the response of thermal conditions, radiation and heat balance of Qinghai Lake to the recent climate change. The results demonstrated that the FLake has a good ability in capturing the seasonal variations of the lake surface temperature and the internal thermal structure of Qinghai Lake. The simulated lake surface temperature showed an increasing trend from 1979 to 2012, positively correlated with the air temperature and the downward longwave radiation, while negatively correlated with the wind speed and downward shortwave radiation. The simulated internal thermodynamic structure revealed that Qinghai Lake is a dimictic lake with two overturn periods occurring in late spring and late autumn. The surface and mean water temperatures of the lake significantly increased from 1979 to 2012, while the bottom temperatures showed no significant trend, even decreased slightly from 1989 to 2012. The warming was the strongest in winter for both lake surface and air temperature. With the warming of the climate, the later ice-on and earlier ice-off trend was simulated in the lake, significantly influences the interannual and seasonal variability of radiation and heat flux. The annual average net shortwave radiation and latent heat flux (LH) both increasing obviously while the net longwave radiation and sensible heat flux (SH) decreasing slightly. Earlier ice-off leads to more energy absorption mainly in the form of shortwave radiation during thawing period, and later ice-on leads to more energy release in the form of longwave radiation, SH and LH during ice formation period. Meanwhile, the lake-air temperature difference increased in both periods due to shortening ice duration.

# 1 Introduction

The Tibetan Plateau (TP) is the highest plateau in the world, known as the Earth's "third pole" (Qiu, 2008), and exerts a significant influence on regional and global atmospheric circulation through its dynamic and thermodynamic effects (Yanai et al., 1992; Duan et al., 2005). The TP is also one of the most sensitive regions to climate change: the surface air temperature
increase over the TP due to global warming is stronger than in other regions (Guo et al., 2012; Duan et al., 2015). Apart from warming, an increase of air humidity and precipitation, and a decrease of shortwave radiation and wind speeds were reported for the central TP since the beginning of the 1980s (Liao et al., 2013; Yang et al., 2014). Thousands of lakes are scattered across the TP, accounting for 39.2 % of the entire number and for 51.4 % of the entire area of Chinese lakes (Ma et al., 2011). Lakes are an inherit components of the hydrological system of TP, named "the world water tower" (Xu et al., 2008),
contributing essentially to the water cycle between atmosphere, glaciers and the major Asian rivers. Due to the significant increase in precipitation and melting of glaciers caused by climate change, the total area of lakes on the TP tended to expand significantly since the late 1990s (Liao et al., 2013; Lei et al., 2014).

Large lake areas significantly influence the local and regional weather and climate, mainly because of their differences in albedo, heat capacity, roughness, and energy exchange compared to the land surfaces around (Bonan et al.,1995; Eerola et al.,
2010). Lakes are very sensitive to climate, and their physical, chemical and biological properties respond rapidly to a climate-related change (Adrian et al., 2009; Williamson et al.,2009). The surface water warming rates of lakes are mainly driven by the increasing air temperature (Adrian et al., 2009; Schmid et al., 2014), depending on combinations of climate and local characteristics, associated with interactions among different climatic factors. Surface water is warming in many lakes around the globe, whereas some lakes are cooling or do not reveal any significant temperature trends (O'Reilly et al.,2015). Global
warming also has an impact on the vertical thermal structure of lakes and cause mixing regimes shifting (Livingstone, 2003, 2008; Boehrer and Schultze, 2008). Surface warming increases the summer vertical stability and prevents the heat transfer to the bottom of the lake, so that a counter-trend of cooling may occur at the bottom (Kirillin et al., 2010). Warming also may result in drastic shifts in the date of lake ice break-up and freeze-up (Weyhenmeyer et al., 2004), which can significantly influence the seasonal thermal and energy regimes of the lakes (Rouse et al., 2003). The ice-on and ice break-up dates on lakes
and rivers demonstrate a long-term trend on later freezing and earlier break-up around the Northern Hemisphere, as a response to the increase in air temperature of about 1.2 °C per 100 years (Magnuson et al., 2000).

Same as globally, both warming and cooling trends occurred in the lakes on TP (Zhang et al., 2014a). Due to the high elevation and low atmospheric density over the TP, the surface received solar radiation input is larger than in lowland areas that results in large diurnal amplitudes of surface temperature (Gao et al., 1981; Ma et al., 2009). During the last decades, a negative trend
in the solar radiation flux was observed over the TP, which can be ascribed to the increase of the air humidity (Shen et al., 2015). As a result, lakes are predicted to experience a cooling trend despite a significant increase of the air temperature over the plateau (Kirillin et al., 2017) that demonstrates decoupling of air and land response to the global change and suggests a non-linear response of the entire hydrological system.

Only a few observation data are available for TP lakes due to the harsh environmental conditions; therefore, the lake thermal conditions and their response to climate change are not well understood. Hence, numerical simulation appears to be the most efficient approach in lake investigation on the TP, provided that the numerical model is well-calibrated and reliable information on the atmospheric forcing is available.

In this paper, we model brackish endorheic Qinghai Lake — the largest lake on the TP and in China — to reveal the major features of the TP lake response to climate change by using the lake model FLake (Mironov., 2008), FLake is a highly parameterized one-dimensional lake model aimed primarily at lake representation in land schemes of regional climate models. The model was numerously tested before for different lakes worldwide (Kirillin, 2010; Bernhardt et al., 2012; Stepanenko et al., 2013; Thiery et al., 2014), including freshwater lakes (Kirillin et al., 2017) and a brackish lake (Lazhu et al, 2016) on the TP. The strength of FLake is its high computational efficiency combined with a realistic representation of the major physics, which made the model to a basic tool for lake representation in the land schemes on the global scale (e.g., Dutra et al., 2010; Salgado and Le Moigne, 2010; Rooney and Bornemann, 2013; Mallard et al., 2014). However, FLake is originally a freshwater lake model taking no account for salinity effects on mixing and heat exchange with the atmosphere. Saline and brackish lakes represent most of the inland water bodies on TP, and they may have an appreciable effect on the land-atmosphere interaction. Therefore, in addition to quantifying the recent climate change effects on the thermal regime of China's largest lake, the second aim of the study is to test the FLake performance on brackish lakes after parameterizations of the salinity effect on the temperature of maximum density and freezing point in the model. Here, we applied the freshwater lake model to a brackish TP lake in order to (i) evaluate the ability of the lake model FLake to simulate the main thermodynamic features of the lake in high-altitude conditions, (ii) validate the performance of a freshwater lake model, extended by simple parameterizations of salinity effects, for a brackish lake.

## 2 Study area, Data, and Methodology

### 2.1 Study area

Qinghai Lake (36°32′-37°15′ N, 99°36′-100°47′ E) is the largest inland lake in China with a surface area of 4 497 km² (in 2017) and a catchment area of 29 660 km². The maximum length and width of the lake are approximately 106 km and 67 km respectively. It is an endorheic, brackish lake (salinity 12.5 g l⁻¹, pH 9.3) (Deng et al., 2010) located on the northeast margin of the TP (Fig. 1) at the height of about 3 194 m a.s.l. The mean and maximum depths of the lake are 21 m and 32.8 m, respectively. The lake is ice-covered from December/January to early April; the average annual lake water temperature is 5.4 °C, with the maximum monthly temperature of 17.2 °C (August) and the minimum of -2.0 °C (January) (Li et al., 2016). The average annual air temperature (1959-2015) at the lake is 1.9 °C. The mean annual precipitation in 1959-2015 was about 340 mm (Ding et al., 2018) with more than 65 % occurring in summer. Annual evaporation from the lake surface was 924 mm, surface runoff water inflow and groundwater inflow were 348 and 138 mm respectively (Li et al., 2007).

Qinghai Lake is sensitive to climate variability: Because the evaporation was generally larger than river runoff and precipitation from 1961 to 2004, the water level of Qinghai Lake decreased at an average rate of 7.6 cm per year (Cui et al., 2016). However, the precipitation continuously increased in 1970-2015 by 15.603 mm per decade according to the data from Gangcha station (the nearest meteorological station approximately 13 km north to Qinghai Lake). Simultaneously, the runoff from the melting of Qilian Mountain glaciers was also increasing because of the regional warming trend of 0.319 °C per decade, coupled with the decreasing evaporation by 1.343 mm per year (observed by Gangcha station) during 1970-2003 (Tang et al., 2018). Since 2004, as the runoff and precipitation exceeded evaporation and the regional climate gradually turned to the direction of "warm and humid", the Qinghai lake level increased at a rate of 14 cm per year during 2004-2012 (Dong and Song, 2011; Zhang et al., 2011, 2014b; Cui et al., 2016).

## 2.2 Data

### 2.2.1 Buoy observation data

The observation data were obtained from the Qinghai Lake hydrological automatic meteorological observation buoy (36.68° N, 100.50° E). The recorded parameters included air temperature, wind direction, wind speed, pressure, relative humidity, surface water temperature at 0.7 m below the surface, dew point temperature and water salinity. The observation period was confined to the summer and autumn open water periods from 2001 to 2005 (Fig. 2), with an observation interval of 3 hours.

### 2.2.2 MODIS Lake Surface Temperature

Some gaps in the long-term buoy observations were caused by harsh environmental conditions and the long ice cover period. Therefore, the 8-day MODIS LST product (MOD11C2), which covers 2001-2012, was additionally used to evaluate the long-term simulated results. This product offers 8-days combined radiative surface temperature approximately at 10:30 and 22:30 local time, which is the satellite transit time, with a resolution of 5 km (Wan et al., 2004). Here we used a single point of MODIS LST closest to the buoy location in order to be comparable with the buoy observed data. We had removed few abnormal values that might be influenced by cloud cover (Langer et al., 2010). The MODIS data comparison against the buoy data found them generally consistent, but MODIS LST was generally lower than buoy observations, with the 2001-2005 average bias of -0.36 °C (Fig. 2). The bias might be attributed to the cool skin phenomenon making the radiation temperature to be typically lower than the bulk temperatures (Robinson et al., 1984; Donlon et al., 2002; Minnett et al., 2003; Leppäranta and Lewis, 2007).

### 2.2.3 Dataset of lake ice phenology in Qinghai Lake

The dataset on lake ice phenology in Qinghai Lake from 2000 to 2018 was built by using RS and GIS technologies based on Terra MODIS surface reflectance product and Landsat TM/ETM+/OLI remote sensing images (Qi et al., 2018). The dataset uses the method of threshold segmentation to extract the ice area of Qinghai Lake based on MOD09GQ product by setting a

reflectance threshold for the red band and a reflectance difference threshold between red and near-infrared bands. The extracted ice area was then validated against the visually interpreted ice area based on Landsat TM/ETM+/OLI images. The dataset includes ice-water vector boundary data, area ratio, and phenological characters in Qinghai Lake from 2000 to 2018. Phenological information includes the start and end dates of lake freeze-up and break-up, and ice cover duration. The dataset provides a reference for exploring the spatio-temporal characteristics of lake ice in Qinghai Lake, as well as for estimating lake ice cover response to climate changes in the region.

### 2.2.4 ITPCAS Forcing Data

The China Meteorological Forcing Dataset (He, 2010) developed by Data Assimilation and Modeling Center for Tibetan Multispheres, the Institute of Tibetan Plateau Research, Chinese Academy of Sciences (hereafter ITPCAS) was used as atmospheric forcing data for the FLake model. The version used here covers the period of 1979-2012. It was produced by merging a variety of data sources, including Princeton meteorological forcing data, Global Land Data Assimilation System (GLDAS) data, The Global Energy and Water Cycle Experiment-Surface Radiation Budget (GEWEX-SRB) shortwave radiation dataset, Tropical Rainfall Measuring Mission (TRMM) satellite precipitation analysis data and China Meteorological Administration (CMA) station data. The ITPCAS forcing data set includes air temperature and specific humidity at 2 m height above the ground, wind speed at 10 m height, surface pressure, precipitation, and downward shortwave and longwave radiations at a spatial resolution of 0.1° and a temporal resolution of 3 hours (He and Yang, 2011). The downward longwave radiation was calculated by the model of Crawford and Duchon's (1999) as a function of air temperature, pressure, specific humidity, and downward shortwave radiation. ITPCAS forcing incorporates CMA station data; therefore, it is more accurate in this region of China compared with other data sets and is generally preferable for modeling studies in China (Chen et al., 2011; Guo and Wang, 2013; Liu and Xie, 2013).

### 2.3 Lake model

The FLake model (Mironov, 2008) is used to simulate the vertical temperature profile and the energy budget of the different layers of the lake on the time scales from several hours to many years. The model divides the lake water body vertically into two layers, the upper layer being the mixed layer with uniform temperature. Beneath the mixed layer, the temperature profile is parameterized using the concept of self-similarity (Kitaigorodskii and Mirokolskii, 1970), which means that the characteristic shape of the temperature profile is conserved irrespective of the depth of this layer. The parameterization formula is:

$$\frac{\theta_s(t) - \theta(z,t)}{\Delta\theta(t)} = \Phi_\theta(\zeta) \quad h(t) \leq z \leq D \tag{1}$$

Where $t$ is time, $z$ is the depth, $\theta_s(t)$ is the temperature of the upper mixed layer of depth $h(t)$, $\Delta\theta(t) = \theta_s(t) - \theta_b(t)$ is the temperature differences across the thermally stratified layer of the depth of $\Delta h(t) = D - h(t)$, $D$ is the lake depth, $\theta_b(t)$ is

the temperature at the lake bottom. $\Phi_\theta(\zeta)$ is a dimensionless "universal" function of the dimensionless depth $\zeta = \frac{z - h(t)}{\Delta h(t)}$ which satisfies the boundary conditions $\Phi_\theta(0) = 0$ and $\Phi_\theta(1) = 1$. Based on the self-similarity assumption, the temperature profile can be expressed as a two-layer approximation:

$$\theta(t) = \begin{cases} \theta_s(t) & 0 \leq z \leq h(t) \\ \theta_s(t) - [\theta_s(t) - \theta_b(t))\Phi_\theta(\zeta)] & h(t) \leq z \leq D \end{cases} \qquad (2)$$

Substitution of Eq. (2) over the lake water column with subsequent substitution into the heat transport equation yields a set of ordinary differential equations, including lake in form of the shape factor $C_\theta = \int_0^1 \Phi_\theta(\zeta)$. The resulting equation system is complemented by an equation for evolution of the mixed layer depth $h(t)$, which is calculated based on the convective entrainment or relaxation-type equation in terms of wind mixing (see Mironov, 2008 for details).

The shape factor $C_\theta$ is parameterized by a relaxation formula:

$$\frac{dC_\theta}{dt} = sign\left(\frac{dh(t)}{dt}\right)\frac{C_\theta^{max} - C_\theta^{min}}{t_{rc}} \qquad C_\theta^{min} \leq C_\theta \leq C_\theta^{max} \qquad (3)$$

Where $t_{rc}$ is the empirically estimated relaxation time (s) of the temperature profile in the thermocline from one limiting curve to the other, following the change of sign in $\frac{dh(t)}{dt}$. $C_\theta^{min} = 0.5$ and $C_\theta^{max} = 0.8$ are the minimum and maximum values of the shape factor.

Additionally, FLake includes the representation of the thermal structure of the ice layer, snow layer and the thermally active upper layer of bottom sediments, all using the self-similarity concept. The snow module of FLake has not been comprehensively tested so far. Compared with other lake models, it is relatively easy to adjust FLake to a specific application due to a small number of lake parameters to be specified, the major ones being the lake depth and the optical characteristics of the lake water.

To partially account for salinity effects in a brackish lake, the freshwater equation of state used by FLake was adjusted by changing temperature of maximum water density ($T_m$) and the freezing point temperature ($T_f$). The parameterization formula of $T_m$ and $T_f$ obtained from linear approximations of empirical function of state of seawater (Caldwell, 1978; UNESCO, 1981) are:

$$T_m[°C] = 3.98 - 0.216S \qquad (4)$$

$$T_f[°C] = -0.055S \qquad (5)$$

Where the S is salinity taken in parts per thousand (‰ or g $l^{-1}$). For the salinity of S=12.5 g $l^{-1}$, which is the case of Qinghai Lake, the equation gives $T_m = 1.28$ °C and $T_f = -0.69$ °C. In addition, the lake depth was set to the mean depth of Qinghai Lake (21 m). The simulation started at the beginning of the year 1979, The forcing data of 1979 were used to drive the Flake model 10 iterations for spin-up. Then the actual modeling period started in 1979 and ended in 2012. The simulation duration was 34

years with the simulation step of 3 hours. The model runs were performed using both original freshwater equation of state and the brackish water approximation (eq. 4-5). Here we defined the simulation with original freshwater equation of state as freshwater lake (FL) experiment and the simulation with the brackish water approximation as saltwater lake (SL) experiment.

## 3 Results

### 3.1 Simulated lake temperatures

Introduction of salinity remarkably affected the ice regime, but not the lake surface temperatures. Therefore, only the simulation results of SL experiment are analyzed in this subsection. Comparison of the simulated lake surface temperatures against MODIS LST (Fig. 3) demonstrated that the FLake model can nicely simulate the seasonal variations of the lake surface temperature: The correlation coefficient amounted at 0.93. The simulated temperature was however generally higher than the MODIS LST, with a positive bias of 1.98°C and an RMSE value of 3.97 °C for annual mean, except in springtime, when the simulated LST had a negative bias of -0.74 °C compared to MODIS LST.

In order to evaluate the effect of the forcing data deviation on the simulation results, we applied a correction to the ITPCAS forcing data. Since the buoy observations are mainly available from June to October, only forcing data for this period of the year was corrected. The air temperature was adjusted with the linear relationship (y = 0.76x + 3.27, x for ITPCAS and y for buoy observations) and the wind speed was corrected by adding a constant bias of 1.19 m s$^{-1}$ between the mean buoy observation and the mean ITPCAS data. After the correction, the bias and root-mean-square error (RMSE) between simulated LST and MODIS LST both reduced for the open water period (from 2.85 °C and 3.71 °C to 2.82 °C and 3.58 °C respectively), especially for the summer (3.30 °C to 1.60 °C) and the autumn (2.97 °C to 1.06 °C). Through the correction of the driving data, we found that positive bias between simulated LST and satellite data can be partly explained by the differences in the forcing weather data measured over the lake and provided by the ITPCAS data. The remaining bias may be partly attributed to the cool skin effect in the LST sensed by MODIS.

The effect of salinity stratification on the lake mixing was not accounted for by the FLake model, assuming purely thermal stratification. The modeled seasonal stratification of Qinghai Lake corresponded to that of a dimictic lake (Fig. 4) with typical features of this type of mixing regime (Kirillin and Shatwell, 2016). Winter and summer stratified periods are divided by two short periods of full vertical mixing (overturns) in late spring and late autumn. During the overturn period, dimictic lakes are supposed to be fully mixed to the bottom. From the simulation, we found that the spring overturn of Qinghai Lake, occurring around May, lasted for 2-3 weeks and the depth of mixed-layer reached the bottom of the lake in most but not all simulation years. The autumn overturn appeared around November-December, lasted approximately for a month, and the mixed-layer reached the bottom of the lake in all simulated years. In the summer stratified period, the mixing process was mainly caused by the wind forcing and the stratification instability due to diurnal temperature variations, and the depth of the mixed layer reached 10-15 m, gradually increasing with time.

Although the modeled LST are slightly lower in the spring and higher in other seasons, especially in summer and autumn, and the deviations in nighttime are larger than in daytime, the model simulated the variations of the LST and its typical magnitudes well, as well as produced a reasonable vertical thermal structure.

**3.2 Response of lake thermal conditions to the long-term trends in external forcing**

According to the ITPCAS data from 1979-2012, the air temperature and longwave radiation had positive trends of 0.58 °C (p<0.01) and 3.22 W m$^{-2}$ (p<0.01) per decade respectively, while the wind speed and shortwave radiation had negative trends of -0.11 m s$^{-1}$ (p>0.05) and -2.41 W m$^{-2}$ (p<0.05) per decade respectively. These values are consistent with other reports on climate change over the TP: The air temperature at different weather stations on TP is rising by an average of 0.09 °C to 0.74 °C per decade from 1961 to 2007 (Guo and Wang, 2012), while, the wind speed and shortwave radiation is decreasing (Yang et al., 2014).

From 1979 to 2012, the simulated LST, mixed-layer temperature, and the lake mean temperature in SL experiment was increasing at 0.74 °C, 0.38 °C and0.26 °C per decade respectively (p<0.01 for all three trends); the bottom temperature revealed a slower trend at 0.2 °C decade$^{-1}$ (p>0.05) (Fig. 5). For the first decade from 1979 to 1989, all temperatures demonstrated a stronger warming trend, especially for the surface layer (1.4 °C decade$^{-1}$, p>0.05). Later on, the trend slowed down to 0.54 °C decade$^{-1}$ (p<0.05) for the surface temperature, same to mixed-layer (0.32 °C decade$^{-1}$, p<0.05) and mean water column (0.14 °C decade$^{-1}$, p>0.05)during the rest of years. The bottom water temperature even demonstrated a slightly decreasing trend of -0.25 °C decade$^{-1}$ (p>0.05).

Due to the importance of the lake surface as an interface of heat and mass exchange between the lake and atmosphere, the relationship between the LST variation trend and main atmospheric characteristics was investigated (Fig. 6). The trend of LST simulated by FLake was consistent with rising air temperature (0.58 °C per decade) but with a higher rate of 0.74 °C per decade. Meanwhile, the simulated LST having a positive correlation coefficient of 0.71 (p<0.01) with air temperature. A negative correlation coefficient of -0.35 (p<0.05) was found between the simulated LST and the wind speed. The downward shortwave radiation had a negative correlation coefficient of -0.29 (p>0.05) to the LST. The LST and the downward longwave radiation were positively correlated (coefficient of 0.74, p<0.01).

**3.3 Lake ice cover**

Compared with FL experiment, the salinity parameterization for $T_m$ and $T_f$ in SL experiment has a certain effect on the ice phenology (Fig. 7): the maximum ice thickness is reduced, the freeze-up date is delayed and the break-up date is advanced, leading to a shorter ice duration period. Nevertheless, the interannual changes between them remained consistent. The simulated freeze-up and break-up date in FL and SL experiments are both later than satellite observations, with some differences in interannual variations but similar range in ice duration. In SL experiment, the maximum ice thickness and the break-up date are closer to the observations, the former was reported of 0.7 m by Chen et al (1995). Hence the ice phenology results from SL experiment were used for further analysis.

The variations of break-up and freeze-up dates are sensitive to the meteorological conditions, e.g. air temperature, solar radiation, and wind (Duguay et al.,2006; Latifovic et al., 2007; Ye et al., 2011; Kirillin et al., 2012; Yao et al., 2016). In SL experiment, the simulated maximum ice thickness demonstrated a negative correlation of -0.52 (p<0.01) to the mean air temperature anomaly from January to April in Qinghai Lake. With the increase of air temperature, the maximum ice thickness of Qinghai Lake reveals a decreasing trend of -0.1 m decade$^{-1}$ (p<0.01) (Fig. 7a). Simulated ice cover of Qinghai Lake started in the late December to January and ended in early April to early May. The correlation coefficient of -0.68 (p<0.01) was found between the freeze-up date and mean air temperature anomaly for November-December (Fig. 7b), while a 0.48 (p<0.01) correlation coefficient was found between break-up date and mean temperature anomaly for March-April (Fig. 7c). With the increasing trend in the air temperature and LST, the freeze-up date delayed about 4.5 (p<0.01) days every decade and the break-up date advanced about 5.7 (p<0.01) days earlier every decade, resulting in shortening of the ice cover period at about 10.2 (p<0.01) days per decade (Fig. 7d).

## 3.4 Interannual variation of energy balance

Lake mean temperature is the indicator of the heat storage in the lake water body, whose changes are mainly driven by the heat exchange at the lake surface. The latter is composed from net (shortwave and longwave) radiation budget, sensible heat flux (SH) and latent heat flux (LH). Quantification of the energy balance at the lake surface is necessary for understanding the mechanisms of the lake response to climate change. The heat transfer from precipitation, runoff and the bottom sediments of the lake are ignored here due to their small magnitudes and observational difficulties.

According to ITPCAS data, the solar radiation flux over Qinghai Lake was decreasing at -2.41 W m$^{-2}$ per decade (p<0.05) (not shown), while simulation results produce a positive trend of 0.78 W m$^{-2}$ (p >0.05) per decade in the net annual shortwave radiation gain by the lake (Fig. 8a). This was caused apparently by shortening of the ice-covered period from 125 days (1979) to 72 days (2012) that reduced the lake surface albedo from the ice values (between 0.1 and 0.6) to the open water albedo (~ 0.07), significantly increasing the amount of net annual solar radiation absorption by the lake. The net longwave radiation reduced at -0.21 W m$^{-2}$ per decade (p>0.05) (Fig. 8b). Concurrently, the downward longwave radiation increased at 3.22 W m$^{-2}$ per decade (p<0.01) (not shown).Hence, the decreasing trend of net longwave radiation was caused by the increased upward longwave radiation (3.4 W m$^{-2}$ per decade, p<0.01) due to the rising LST during the open water period and shortening of the ice cover duration.

From 1979 to 2012, SH at Qinghai lake decreased slightly at -0.14 W m$^{-2}$ per decade (p>0.05) (Fig. 8c), while LH become stronger at 1.57 W m$^{-2}$ per decade (p<0.05) (Fig. 8d). Hence, the additional heat gained due to the net radiation increase (0.57 W m$^{-2}$ per decade, p>0.05) was mainly balanced by the increased LH due to evaporation. The average annual energy storage in the water body (Qs, ice cover not included) of Qinghai Lake was close to equilibrium and showed a slight downward trend (-0.58 W m$^{-2}$ per decade, p>0.05). The long-term inter-annual cumulative energy storage in turn showed an increasing trend (4.68 W m$^{-2}$ per decade, p<0.01), which was consistent with the increasing lake mean water temperature.

## 3.5 The lake-air temperature difference and the radiation flux

The strong seasonal variation of the surface-air temperature differences is driven by the different thermal properties of the lake and the surrounding land surface (Haginoya et al., 2009; Desai et al., 2009). In the seasonal course averaged over the period 1979-2012, the 5-day moving average air temperature was higher than LST during the ice-covered period (Fig. 9a), with the minimum lake-air temperature differences as low as -6.9 °C (Fig. 9b). After the ice-off between early April to mid-May, LST increased rapidly, particularly due to heating by the intense solar radiation (maximum 285.4 W m$^{-2}$ for 5-day average), characteristic for high-altitude conditions on TP (Fig. 9c), and exceeded the air temperature in June, reaching the maximum of 18.7 °C in August (Fig. 9a). LST was generally higher than air temperature from June to January of the next year, which roughly coincided with the end of the open-water period. The mean lake-air temperature difference (5-day moving average) became positive in June and kept increasing to a maximum of 12.8 °C in December just about 20-30 days before the ice cover formation (Fig. 9b). Owing to the large heat capacity of the lake, a clear phase lag existed between LST and air temperature. The time difference between the seasonal temperature maximums of the LST and the air temperature was about 20 days, and the time difference between both values dropping to the freezing temperature of water of 0 °C was about 2 months.

From the perspective of interannual variability, both air temperature and LST had an increasing trend all year round, with stronger warming in winter than in summer (Fig. 9a). Although the downward longwave radiation increased in summer and autumn (average of 0.33 W m$^2$ a$^{-1}$, maximum of 0.47 W m$^2$ a$^{-1}$ in September) (Fig. 9d), the LST increased slower than the air temperature resulting in a reduction of lake-air temperature difference in autumn (average of -0.04 °C a$^{-1}$, maximum of -0.05 °C a$^{-1}$ in November) (Fig. 9b). This behaviour can be attributed to the apparent decrease of the downward shortwave radiation in summer (average of -0.59 W m$^2$ a$^{-1}$) and in autumn (average of -0.11 W m$^2$ a$^{-1}$), which reduced the net shortwave radiation absorption by the lake in summer and autumn (average of -0.55 W m$^2$ a$^{-1}$ and -0.10 W m$^2$ a$^{-1}$ respectively, Fig. 9c, e). In turn, the increased upward longwave radiation in summer and autumn (average of ~ 0.22 W m$^2$ a$^{-1}$) (not shown) partially damped the effect of downward longwave radiation increase (average of 0.33 W m$^2$ a$^{-1}$), which lead to a decrease of the net longwave radiation from the lake to air (average of ~ 0.19 W m$^2$ a$^{-1}$) between mid-summer and late-autumn (Fig. 9f).

In contrast to the depressed trend of LST increase during the ice-free period, the monthly mean LST in early winter and late spring increased more rapidly than the air temperatures, with two apparent peaks of 0.24 °C a$^{-1}$ and 0.12 °C a$^{-1}$ in January and May, respectively (Fig. 9a, b). The significant increase of LST in these periods may be related to the shift of the ice-on and break-up dates (grey areas in Fig. 9), as well as to the slight increase of solar radiation in December and April. The same seasonal pattern was reflected in the variations of the radiation balance (Fig. 9e, f): During the open water period, the absorbed (released) net shortwave (longwave) radiation of lake water had a generally consistent trend with downward shortwave (longwave) radiation, while during the ice formation period (the grey area around December-January in Fig. 9) and the thawing period (the grey area around April-May in Fig. 9), the absorbed (released) net shortwave (longwave) fluxes had opposite trends to the downward shortwave (longwave) radiation fluxes. The net shortwave radiation increased obviously in these two periods (average of 0.57 W m$^2$ a$^{-1}$ and 0.53 W m$^2$ a$^{-1}$ respectively) while the downward shortwave radiation was not (average of -0.05

W $m^2$ $a^{-1}$ and -0.13 W $m^2$ $a^{-1}$ respectively) (Fig. 9c, e). The same is true for the net longwave radiation that decreased obviously in these two periods (-0.31 W $m^2$ $a^{-1}$ and -0.40 W $m^2$ $a^{-1}$ respectively) although the downward longwave radiation had an increasing trend in each period (0.34 W $m^2$ $a^{-1}$ and 0.16 W $m^2$ $a^{-1}$) (Fig. 9d, f).

## 3.6 Heat budget during ice-on and ice-off

Since the dates of the lake ice break-up and freeze-up strongly affect the seasonal energy budget of the lake (Rouse et al.,2003; Jakkila et al., 2009), the heat budget and its long-term trends were considered in more details. During the thawing period (the grey area around April-May in Fig. 10, same as in Fig. 9), the solar radiation was the strongest (average of 270.6 W $m^{-2}$, Fig. 9c), on the background of appreciable downward longwave radiation (average of 246.1 W $m^{-2}$, Fig. 9d). An earlier break-up date significantly reduced the albedo of the lake (from ice ~ 0.6 to water ~ 0.07), which increased the net shortwave radiation

into the lake (Fig. 9e), same for the net radiation in April (1.1 W $m^{-2}$ $a^{-1}$, Fig. 10c). Concurrently, the small lake-air temperature differences also ensured small SH and LH (average of -3.3 and 10.4 W $m^{-2}$ respectively, Fig. 10a, b), which means the heat release from the lake surface (the compound of net longwave radiation, SH, and LH) was low (Fig. 10d). As a consequence, the energy storage (Qs) of the lake water body in this period (average of ~ 91.7 W $m^{-2}$) increased due to earlier ice breakup at ~1.0 W $m^{-2}$ $a^{-1}$ or at ~ 1.1% per year (Fig. 10e).

In contrast, the freezing period (the grey area around December-January in Fig. 10) was characterized by the weakest levels of both downward shortwave and downward longwave radiation (average of ~125.9 W $m^{-2}$ and ~166.5 W $m^{-2}$ respectively, Fig. 9c, d). However, the upward longwave radiation (not shown) in this period was ~1.7 times larger than downward longwave radiation, causing a minimum of net radiation of -45.3 W $m^{-2}$ in December (Fig. 10c). Also, unlike in the thawing period, the values of upward SH and LH (average of 20.4 W $m^{-2}$ and 26.9 W $m^{-2}$ respectively) both cannot be ignored because of a large

lake-air temperature difference during this period. Hence, a later ice-on leads to an intense cooling of the lake water because the upward longwave radiation, SH and LH were not obstructed by the ice cover (Fig. 10 a, b, c, d). The additional heat absorbed by the lake caused by earlier break-up in previous seasons released before freeze-up partly contributed to the delay of the freeze-up date of Qinghai Lake.

## 4 Discussions and Conclusions

**4.1 Model performance**

The validation results indicate that FLake performed well for the extreme climatic conditions of TP. Although there is an underestimation to the lake surface temperature in spring and an overestimation of it in rest of the seasons, it reproduced well the observed seasonal variation of LST and simulated reasonably the thermal structure of the mixed layer and the thermocline. FLake can be considered a useful tool to study the impact of the climate change to lakes on TP.

The ITPCAS forcing data incorporating observations from land weather stations, produced a constant bias when applied directly to model the lake surface conditions. The reason is the difference in the physical characteristics, in particular, air

temperatures and wind speeds, between land and water, which is especially strong over TP (Lazhu et al., 2016). The result is consistent with the findings of Kheyrollah Pour et al (2012), who applied the FLake model to Great Slave Lake and Great Bear Lake in Canada. They also found that the model overestimated the LST compared with the MODIS data because the forcing data were obtained at the land station rather than over the lake surface. In our case, the ITPCAS air temperatures are 0.71 °C

larger during daytime and 2.49 °C smaller during nighttime compared to the buoy observations in summer and autumn. The ITPCAS wind speeds are 0.63 m s$^{-1}$ lower during daytime and 1.83 m s$^{-1}$ lower during nighttime. In turn, the daily variations of the air temperatures in ITPCAS data are almost 2.85 times higher than in the buoy observations. Lower wind speeds from ITPCAS forcing data weaken the heat transfer and leads to a warmer lake surface temperature simulated by FLake. Additionally, the wind speeds of ITPCAS data are smaller at night corresponding with a higher simulated LST in the nighttime.

This result proves that the deviation of the ITPCAS forcing data indeed leads to a warmer simulated LST. Hence, the choice of atmospheric forcing is crucial for the simulation of large lakes in the extreme highland conditions of TP. In the absence of long-term weather observations over the lake surface, a correction procedure can be applied to the forcing data based on the available short-term observations from moored stations (buoys) and (or) satellite information. In this study, a comparison with a short-term observation data from the buoy on lake surface allowed correction of the ITPCAS forcing data significantly

reducing the bias between the model and the remote sensing data.

The simulated seasonal stratification regime suggests that Qinghai Lake is dimictic, with the spring overturn taking place around May and the autumn overturn appearing around November-December. Currently, there is no long-term information available on the vertical thermal structure of Qinghai Lake. The stratification pattern simulated in this study is however very similar to the observations from other Tibetan lake, Bangong Co (Wang et al, 2014). Salinity can influence the temperature of

maximum density ($T_m$) and the freezing temperature of water ($T_f$). According to the 12.5g l$^{-1}$ salinity of Qinghai Lake, these two parameters equal to 1.28 °C and -0.69 °C instead of the default model configurations of 4 °C and 0 °C respectively. Considerations of the salinity effects lead to a slightly earlier spring overturn and a later autumn overturn, and consequently to an extension of the lake stratification period. Because the salinity stratification effects cannot be completely included in the model designed for freshwater lakes, the simulated mixing regime may have some differences from the actual situation of

Qinghai Lake.

Despite incorporation of the salinity effects on $T_m$ and $T_f$ improved simulation accuracy of maximum ice thickness and break-up date, the ice phenology modeled by FLake still differs from the remote sensing observations. The discrepancy may be related to a number of factors not included in the model. One of them is the effect of salinity on the ice structure, density, and porosity; the others are precipitation, inflows, circulation under ice cover and wind, which is especially important for large-

area lakes (Kirillin et al. 2012), such as Qinghai Lake. However, the air temperature apparently has the strongest effect on ice regime, especially in long-term changes, which appear to be well-simulated by FLake allowing us to study the effect of climate change on lake ice regime within the model ability.

Following the studies of Lazhu et al. (2016) on Nam Co Lake (salinity ~1.78 g l$^{-1}$) and Kirillin et al. (2017) on freshwater Lakes Ngoring and Gyaring, the good prediction of the LST over the largest, brackish lake of TP by the relatively simple,

highly parameterized model FLake, verified by satellite and buoy data, is one of the core results of this study. Both the importance of the TP for global climate interactions and the lack of continuous observations in this region demand reliable modeling schemes to take into account the complexity of the land-atmosphere interactions. FLake is currently among the few lake parameterization schemes actively used in regional climate models and numerical weather prediction (NWP).

Complementary to the recent study of Kirillin et al. (2017), who successfully applied FLake to the freshwater lakes of TP, the present study demonstrates that the model adequately simulates the major mechanisms of the air-lake interaction in large brackish lakes of TP. Hence, FLake can significantly improve the simulation of the land-atmosphere interaction in regional climate models and NWP, which is crucial for understanding the climate-driven changes in this key region. To a first approximation, the result suggests the applicability of FLake to the simulation of all large brackish waters. The latter are

characteristic features of arid regions worldwide having a strong impact on regional climate and water budget.

We have found that the duration of the ice-covered period is crucial for the lake-atmosphere interaction on the TP, with periods of ice-on and ice-off having the strongest effect both on the radiation balance and the boundary heat exchange by SH and LH. To simulate the ice cover duration properly, the heat storage in winter and the vertical heat transport across the ice-covered water column should be adequately described. In its present version, FLake treats these in a simplistic way, neglecting the

heating of water column by solar radiation penetrating the ice cover (Kirillin et al., 2017). This simplification is a source of potential errors in the simulated ice break-up date and LST after the break-up, which errors can be significant for the Tibetan conditions, taking into account the strong solar radiation and low snow precipitation on TP in winter. In earlier studies on lowland lakes, FLake tended to predict earlier break-up dates because of the absence of snow in the FLake model (Bernhardt et al., 2012; Kheyrollah Pour et al., 2012). In this study, the simulated break-up date is generally later than observation, which

can be treated as an indication of the importance of the under-ice water column heating by solar radiation neglected in the model. Another factor potentially introducing the uncertainty into the simulation of the ice duration is the ice albedo. The latter was recently estimated in Qinghai Lake to be much lower than typical estimates for lakes: ice albedo obtained by MODIS was less than 0.25 under the snow-free condition and less than 0.4 even under the snow cover condition (Li et al., 2018, Lang et al., 2018). Among the reasons for such a low ice albedo may be mentioned is the effects of salt on the ice structure and

deformation of the ice surface under the influence of the strong solar radiation. As a result, standard modeling approaches may underestimate the amount of shortwave radiation penetrating the ice, with subsequent errors predicting the ice duration and underestimation of the LST after ice break-up date.

The LST acquired by the MODIS, which is used as a reference for validation of simulation results, is generally lower than the in situ LST. This discrepancy may partly be contributed by the cool skin effect (Crosman and Horel, 2009), which is also found

to be stronger in high-altitude lakes than in the ocean due to strong solar radiative heating and cooler air temperature at lake surface (Li et al., 2015; Wen et al., 2016). This suggest that the model predictions of the bulk LST may be better than comparison against the satellite data shows, though exact estimation and correction of the cool skin effect is out of the scope of this study.

## 4.2 Response of Qinghai Lake to climate change

As expected, the correlation analysis shows that the changes in LST are closely related to air temperatures, downward longwave radiation and wind speed (Fig. 6). The increase of the air temperature and downward longwave radiation plays a key role in lake surface temperature warming, and the decrease in wind speed also promoted the warming of the lake surface temperature. The downward shortwave radiation is negatively and insignificantly (R<0.3) correlated with the water temperature that also can explain the slower increase of the water temperature compared with the air temperature (see Kirillin et al. 2017). The decrease in ice cover duration increases in turn the annual amount of shortwave radiation penetrating to the water column, accelerating the net warming. Annual mean LST simulated by FLake increased at a rate of 0.66 °C per decade, primarily due to rising air temperature and decreasing wind speed. The warming trend of simulated LST significantly exceeded that of the regional air temperature (0.58 °C decade$^{-1}$). This discrepancy may be caused by declining winter ice cover, which leads to an earlier start of the stratified season that significantly increases the LST (Austin, 2007). Mixed-layer and water mean column temperature increasing 0.38 °C and 0.26 °C from 1979 to 2012 respectively, while the bottom temperature increased slowly from 1979 to 1989 and has even a slight decrease trend from 1989 to 2012. The slight decrease of the deep temperatures agrees with findings of Kirillin et al (2017) from freshwater Ngoring Lake in TP, and the research of Huang et al. (2017) that use the GLM lake model at another TP lake Nam Co. The apparent reason for the deep cooling is the increase of stability of the lake due to surface warming, which restricts heat transfer from surface to bottom and produces a decrease of the bottom water temperature. This behaviour has been reported as a characteristic in previous studies on lowland dimictic lakes (Hondzo and Stefan,1993; Danis et al.,2004; Kirillin, 2010).

As mentioned above, climate change is found to have a strong impact on lake ice phenology. The maximum ice thickness of Qinghai Lake is decreasing in simulations, significant tendencies to later ice-on and earlier ice-off are predicted. These three ice phenology characteristics are correlated with the January-April, November-December, and March-April air temperature respectively. The ability to accurately represent ice cover on lakes is essential for the improvement of global circulation models, regional climate models and numerical weather forecasting (Brown, 2010). We have shown that the net shortwave radiation increase caused by shortening of the ice duration plays a key role in net radiation increase. Hence, the declining winter ice cover has a significant influence on annual radiation balance of the lake.

In total, the annual energy storage in water body of Qinghai Lake (Qs) decreased at a slow rate of -0.58 W m$^{-2}$ per decade, influenced primarily by the increase of received net radiation and released LH at the lake surface (Fig. 8 e). Still, the cumulative energy storage of the lake is increasing at 4.68 W m$^{-2}$ per decade (p>0.01) (Fig. 8 f), consistent with the trend of the mean water column temperature. The change of freeze-up/break-up date dramatically influenced the lake energy and heat budget during the ice formation/decay period. The earlier thaw of ice causes an increase of energy absorbed by the lake in late spring since more solar radiation comes into the lake without reflection by the ice cover. The delayed freeze-up date leads to an increase in energy lost before freeze-up due to strong upward longwave radiation, SH and LH.

## 4.3 Differences between the highland TP lakes and lakes of other regions

For low-altitude temperate and boreal lakes, the air temperatures are typically higher than LST after the ice-off and remain higher until temperature equilibrates around mid-summer. In the subsequent period down to ice-on, the LSTs are typically higher than the air temperatures. Hence, the atmospheric boundary layer is generally stable throughout much of the summer season over low-altitude lakes (Scott and Huff, 1996; Rouse et al., 2003; Gianniou and Antonopoulos, 2007; Momii and Ito, 2008; Nordbo et al., 2011). Due to a higher altitude, the lakes on the TP have a lower atmospheric thickness and air density, the solar radiation over the plateau is much stronger than in other areas of the same latitude, while the air temperatures are comparably low (Wen et al., 2016; Haginoya et al., 2009; Li et al., 2016). These specific climatic conditions cause a significantly different seasonal interaction between the lake and the atmosphere. In this study, the LST of Qinghai lake increased very fast after ice melt in mid-April under the strong solar radiation and equilibrated with air temperature in June, which is much earlier than in low-altitude lakes. The difference between air temperature and LST is the fundamental property of lake-air interaction, determining the intensity of the surface heat exchange by means of atmospheric stability. When the LST is higher than air temperature, which is the case of TP lakes in summer, the atmosphere over the lake becomes increasingly unstable, accelerating the release of heat to the atmosphere by convection. In that sense, the role of lakes, as hot-spots of the land-atmosphere interaction on the TP, consists in the accumulation of the solar radiation and release of the accumulated heat into the air by the convective exchange. This fact determines also the differences in the response of TP lakes to regional climate change compared to that found previously in low-altitude areas.

## Data availability

ITPCAS dataset is available in the Third Pole Environment Database (http://en.tpedatabase.cn/portal/). The dataset of ice phenology in Qinghai Lake from 2000 to 2018 is available in the Chinese Scientific Data (http://www.csdata.org/p/214/). The lake model FLake is available from the model community site (http://www.lakemodel.net). The model configuration files and the output of the lake model are available from the first author by request.

## Author contribution

D. Su and L. Wen conceived the study. X. Hu provided the buoy data. D. Su performed the modeling with contributions from L. Wen and G. Kirillin. L. Zhao, Z. Li and J. Du performed analysis of remote sensing data. D. Su, L. Wen, S. Lyu, X. Gao and G. Kirillin analyzed the model output. D. Su wrote the paper with contributions from all co-authors.

## Competing interests

The authors declare that they have no conflict of interest.

## Acknowledgements

The study was supported by the National Natural Science Foundation of China (NSFC, 91637107), the bilateral research project GZ1259 supported by the Sino-German Center for Research Support, CAS "Light of West China" Program Y929641001 and NSFC 41775016, 41605011, 41811530387. G. Kirillin was supported by the German Science Foundation (DFG Projects KI-853-7/1, KI-853-11/1). Authors are grateful to M. Leppäranta for advice and comments on the paper.

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

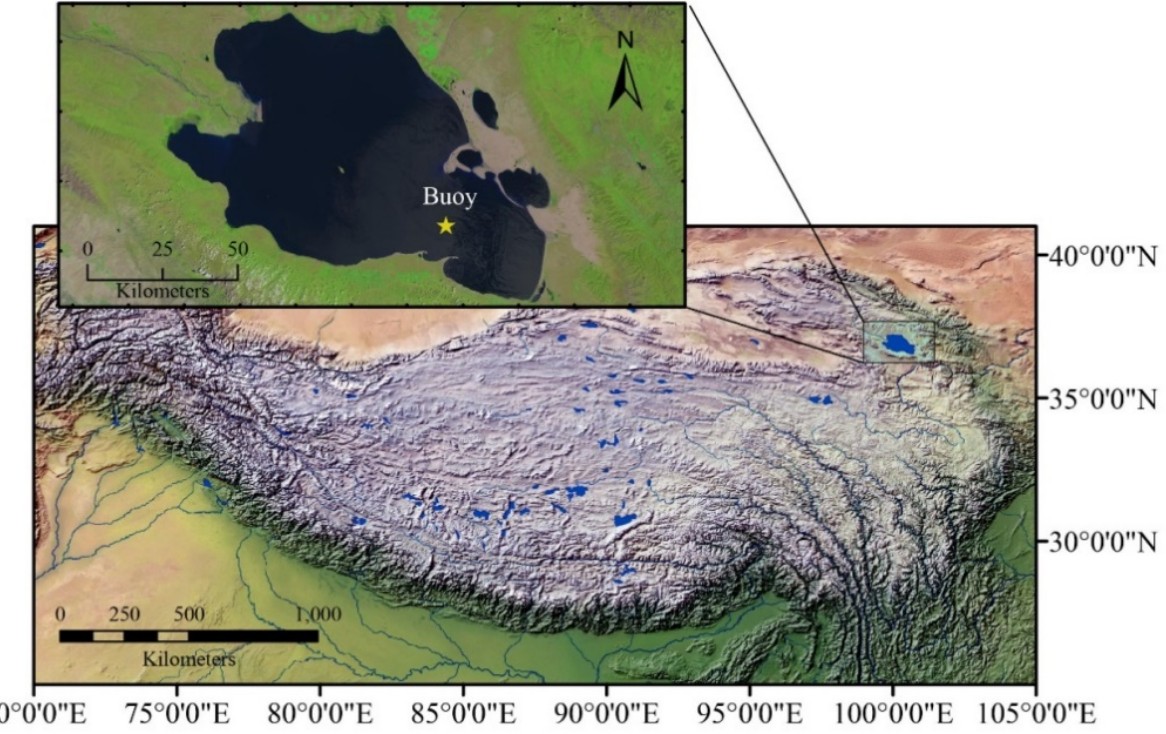

**Figure 1: Study area and the location of the buoy station.**

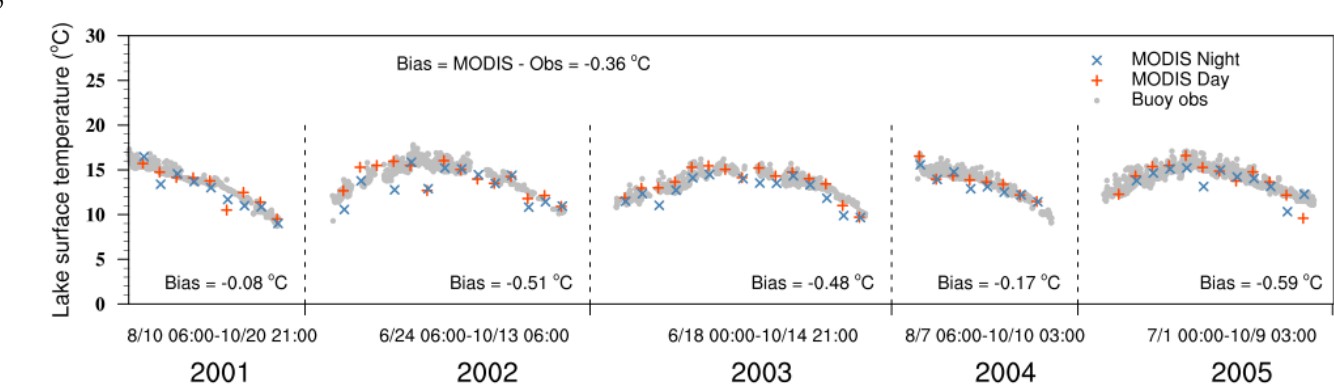

**Figure 2: Comparison of the lake surface temperature between observations from buoy and MODIS respectively.**

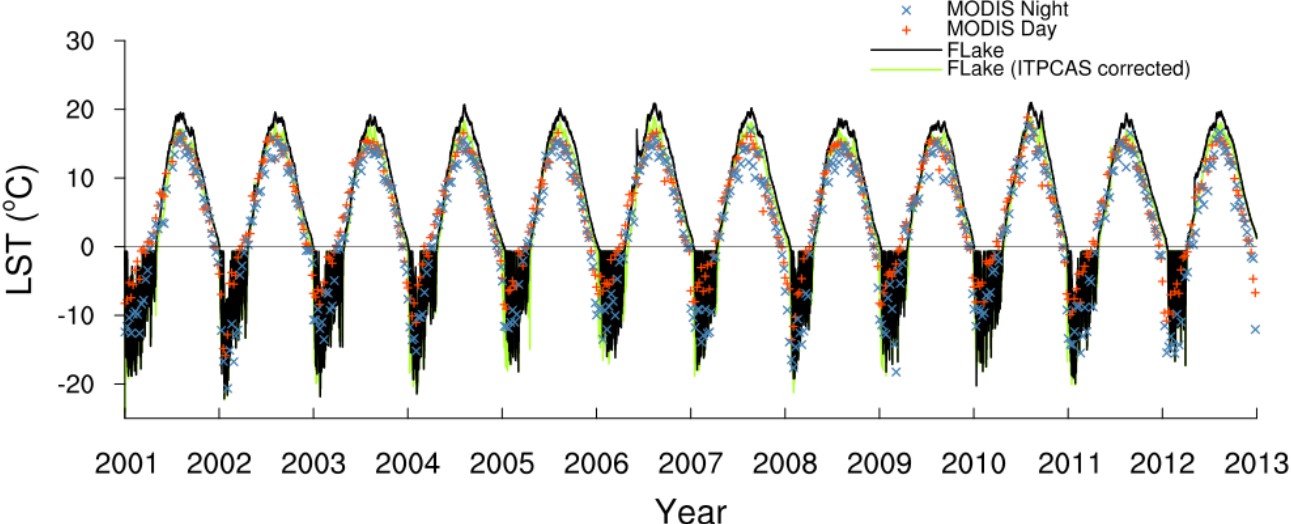

**Figure 3: Comparison of lake surface temperature (LST) between FLake simulation forced by original (black line) and corrected (green line) ITPCAS data, and MODIS observation (cross markers).**

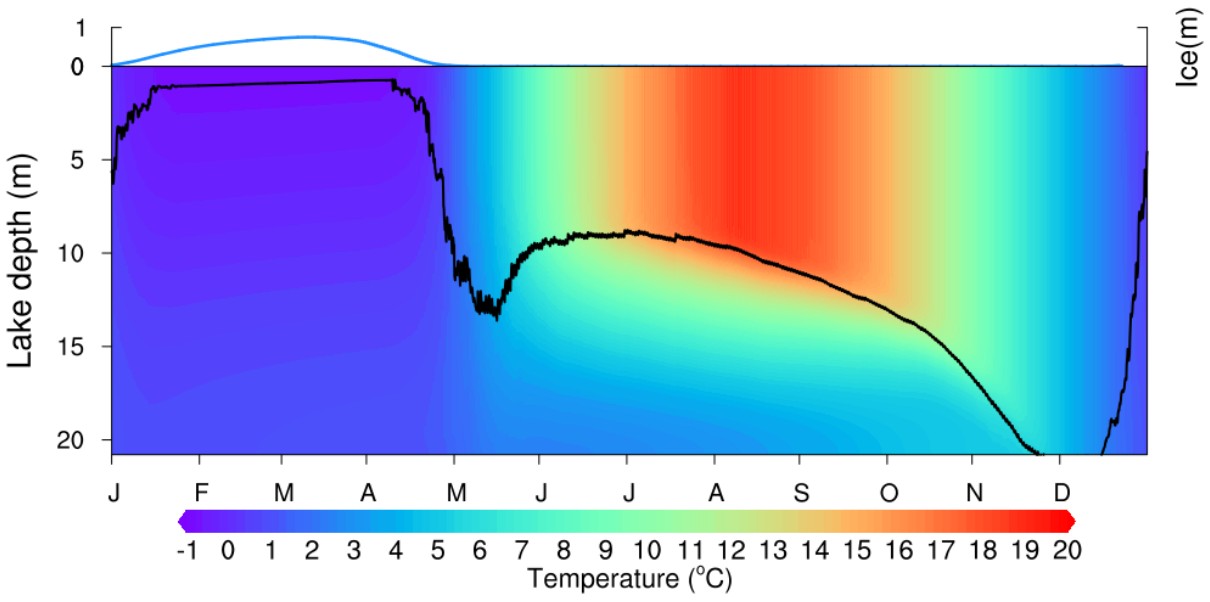

**Figure 4: The modeled seasonal thermal stratification pattern and ice cover of Qinghai Lake averaged from 1979 to 2012. The blue line is ice cover thickness and the black one is the depth of the mixed layer.**

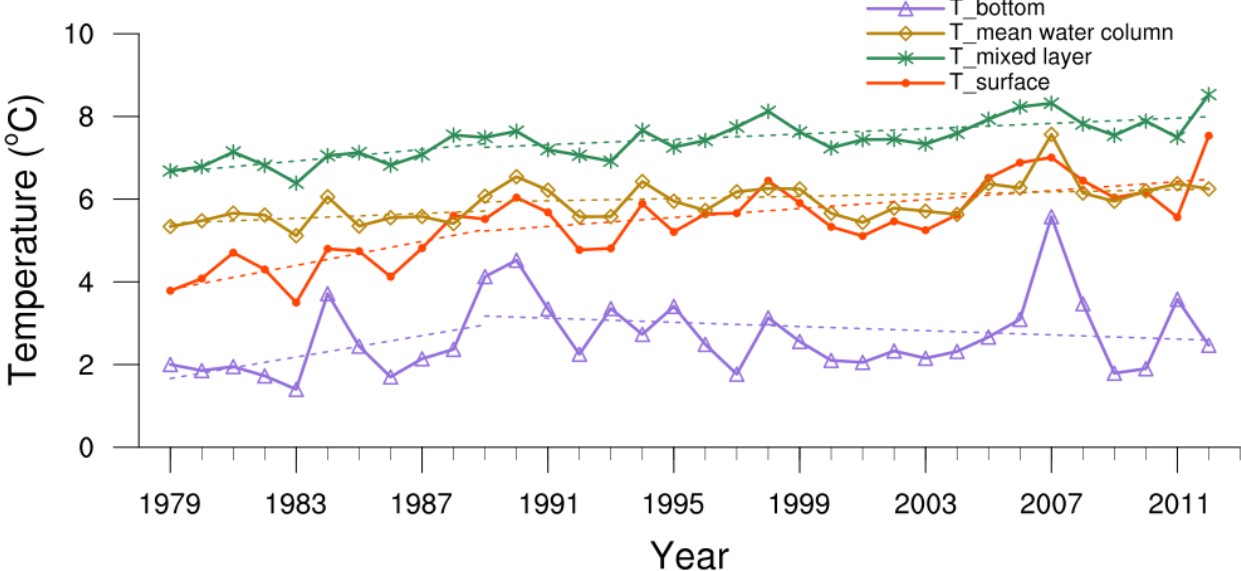

**Figure 5: Annual variation trends of the lake water temperature at the surface, mixed-layer, mean water column and bottom layer respectively.**

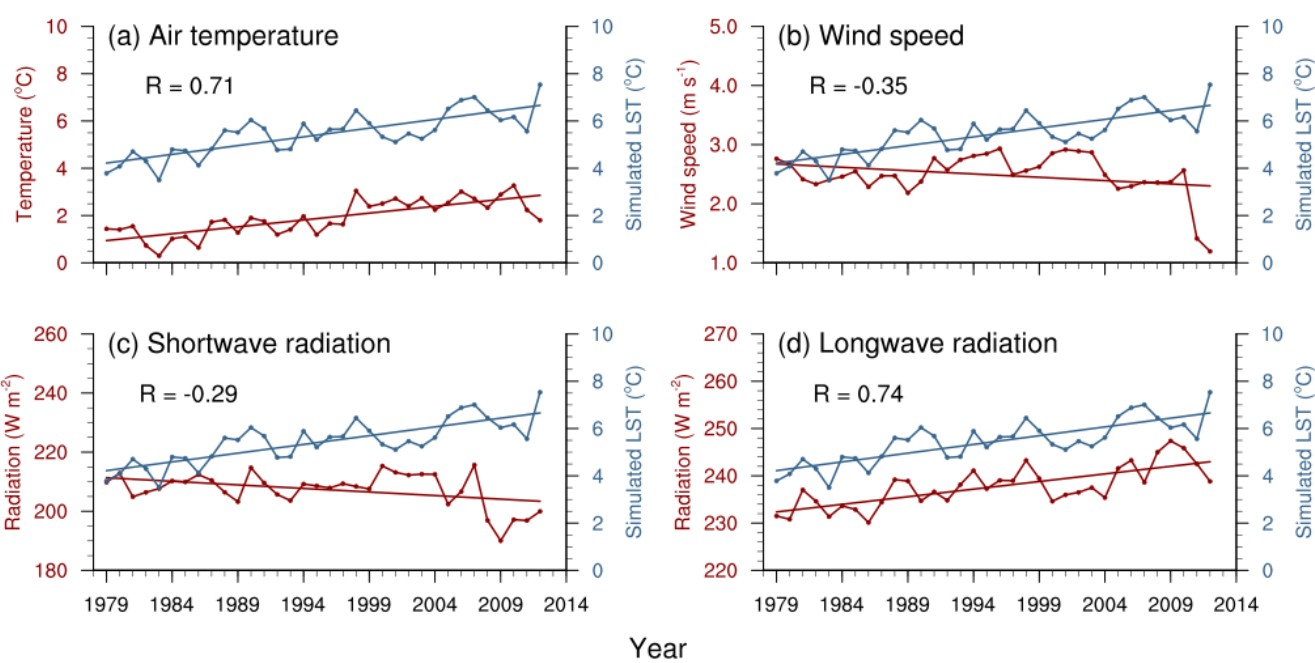

**Figure 6: Interannual variations of annual air temperature (a), wind speed (b), shortwave radiation(c) and longwave radiation (d) at Qinghai Lake and their correlations with simulated annual mean lake surface temperature (LST) from 1979 to 2012.**

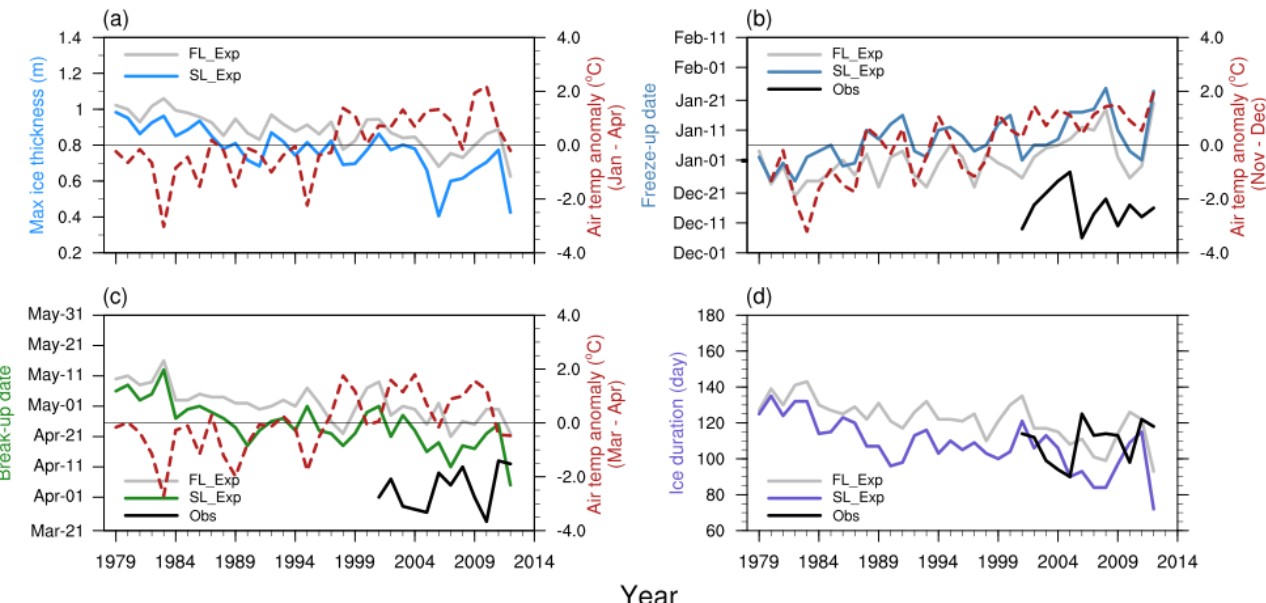

**Figure 7: The interannual variations of simulated annual maximum ice thickness (a), freeze-up date (b), break-up date (c) and ice duration (d) of Qinghai Lake. The coloured (grey) line indicate SL (FL) experiment. The red dash line is air temperature anomaly in the specified period and the black line is ice phenology observation derived from the satellite.**

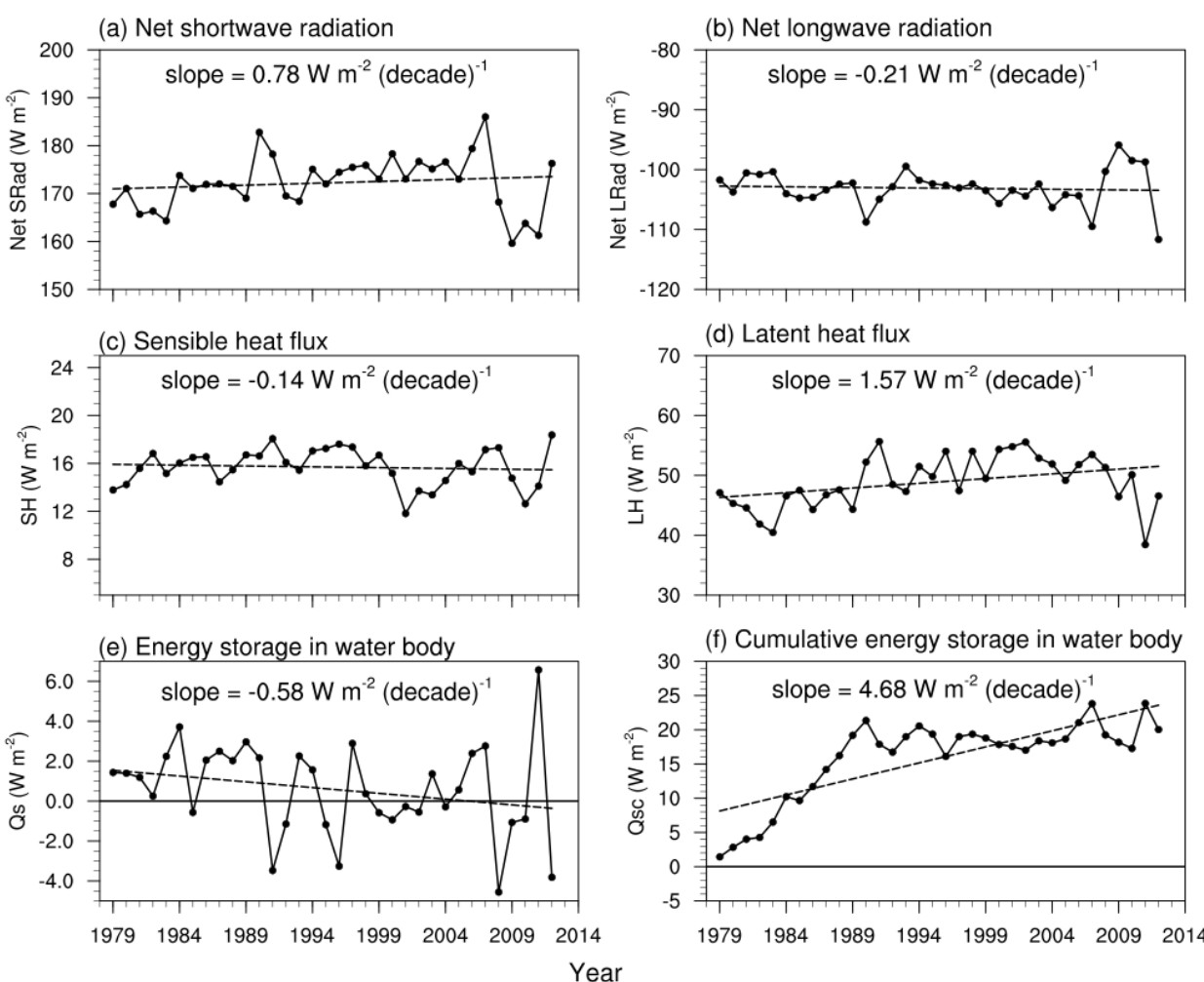

**Figure 8: The interannual variation trend of simulated annual mean lake surface net shortwave radiation (a), net longwave radiation(b), sensible heat flux(c), latent heat flux (d), energy storage in water body (e) and cumulative energy storage in water body (f) from 1979 to 2012.**

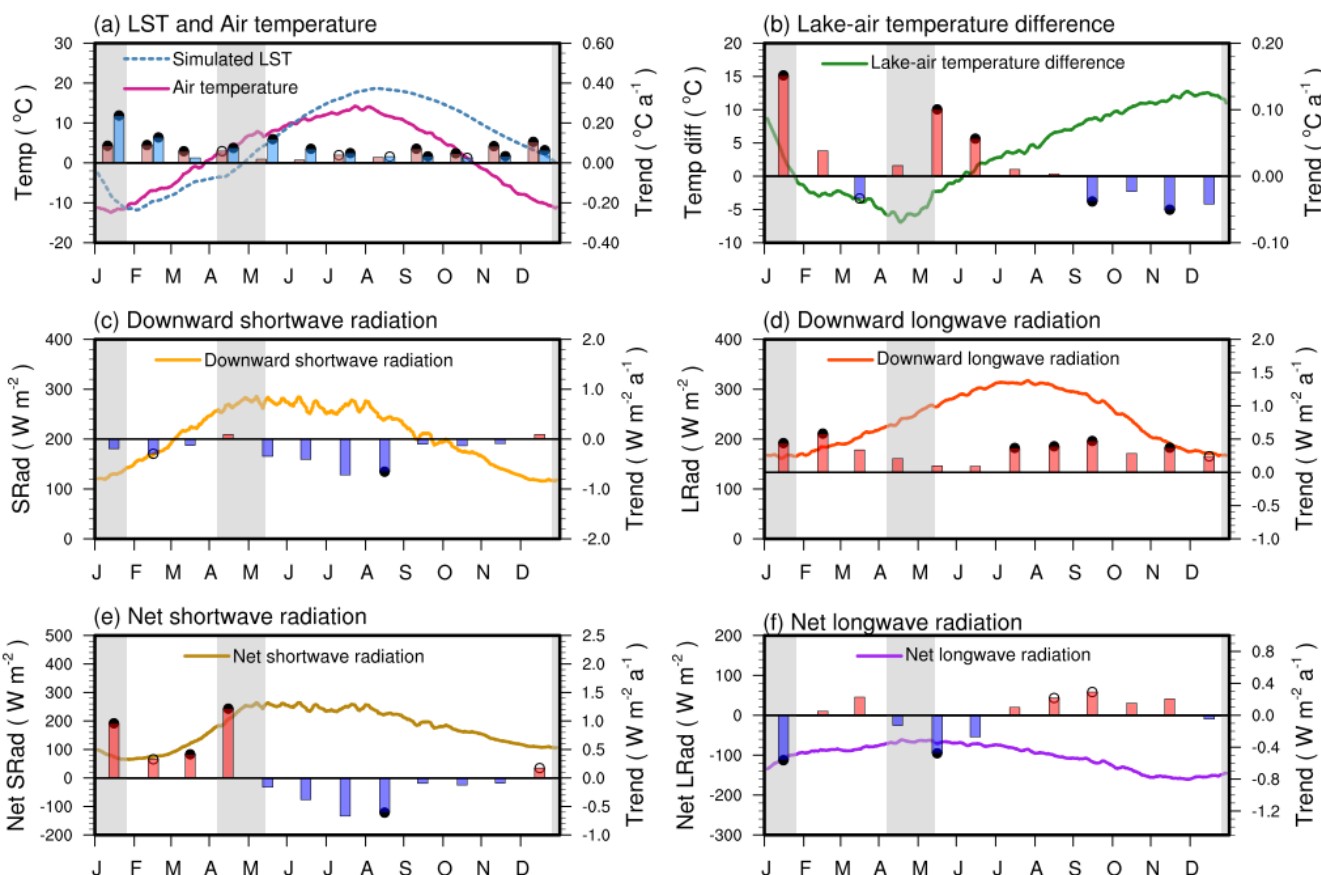

**Figure 9: Climatological mean seasonal variations (5-day moving average, lines) in simulated LST and air temperature (a) with their difference (b), downward shortwave radiation (c), downward longwave radiation(d), net shortwave radiation (e) and net longwave radiation (f) at lake surface. The bars indicate their monthly averaged mean annual variation trend from 1979 to 2012, red for positive and blue for negative except in (a) that for air temperature and LST respectively. Solid points at end of the bars mean pass the significance test of p<0.01 and hollow points mean p<0.05. The grey areas indicate the freeze-up and break-up date variation range of the lake.**

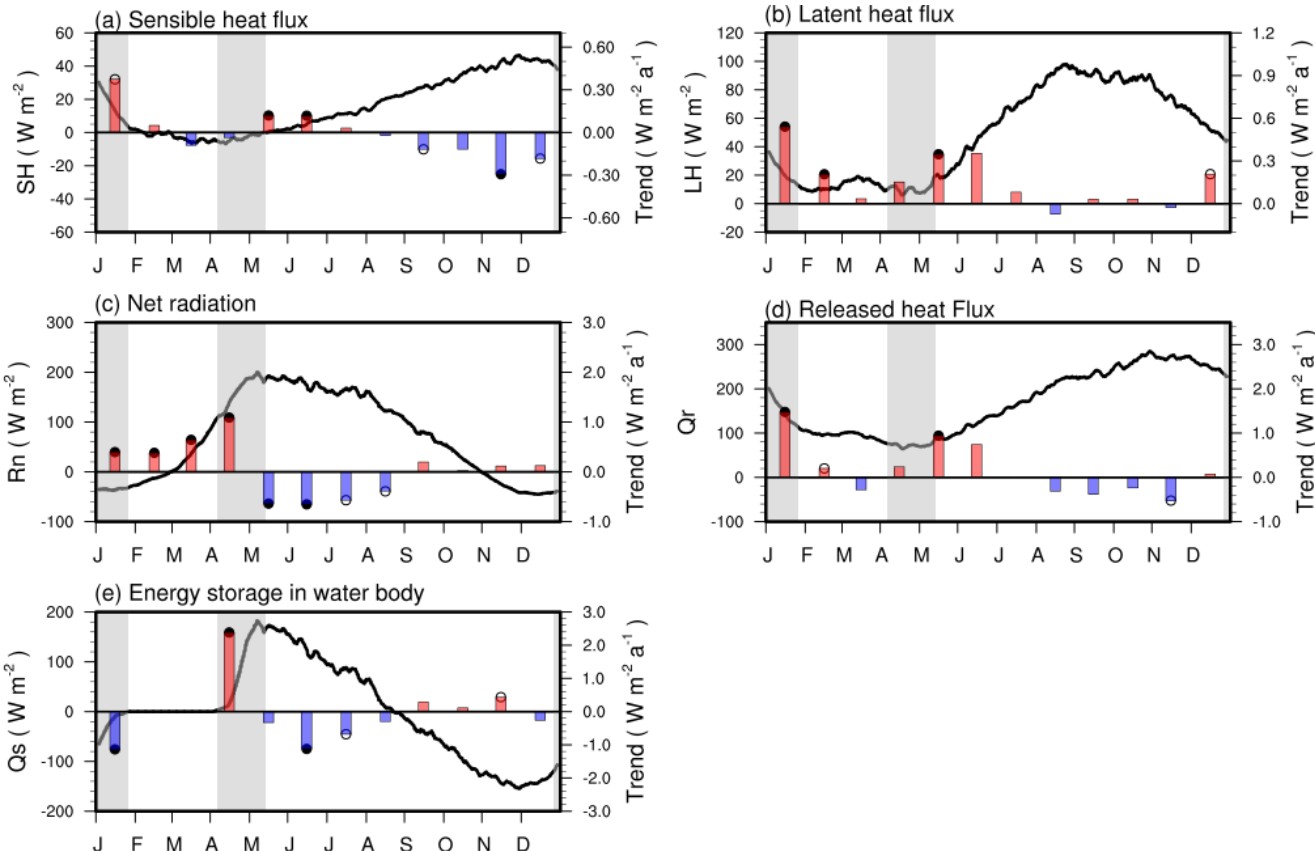

**Figure 10: Climatological mean seasonal variations (5-day moving average, lines) in the simulated sensible heat flux (a), latent heat flux (b), net radiation (c), released heat flux at lake surface (d) and energy storage in water body (e). The bars indicate their monthly averaged mean annual variation trend from 1979–2012, red for positive and blue for negative. Solid points at end of the bars mean pass significance test of p<0.01 and hollow points mean p<0.05. The grey areas indicate the freeze-up and break-up date variation range of the lake.**