# Peer review of "Numerical study on the response of the largest lake in China to climate change"

_Hydrology and Earth System Sciences, 2018_

## Referee Comment (RC1) · Anonymous Referee #1 · 4 Jan 2019

The paper is focused on Qinghai Lake, the largest of thousands of lakes situated on Tibetan Plateau, China. The lake is brackish with salinity about 12.5 g/L. The authors use the well-known one-dimensional model FLAKE forced with a local set of historical gridded meteorological data for the period 1979-2012 to simulate the thermal and ice regimes and their ongoing trends accompanying the global warming. Because the Qinghai, as well as all other Tibetan lakes, has been very sparsely covered by in situ measurements, and virtually no field monitoring data are available (except those from a single meteorological buoy used in this study), numerical simulation is the only mean capable of giving quantitative insights into the long-term variability of the Tibetan lakes. Therefore, in my opinion, the article presents interesting and useful information and should be published after moderate revision.

My only general comment about this manuscript is as follows: I think that the possible role of salinity and its changes in the estimated long-term variability of the lake regime should be evaluated and discussed more thoroughly. For instance, can the trends of the ice regime (Section 3.3) be associated not only with the air temperature increase, but also, at least partly, with salinity increase over the period 1961-2004? According to the information supplied in Section 2.1, the lake level dropped for about 3.3 m during this period, which, given the mean depth of 21 m and mean salinity about 10 g/L, implies salinity increase of about 2 g/L. This, in turn, may have affected the ice regime.

Generally, salinity may exercise influence on the issues addressed in the article through (1) salinity stratification, which is not accounted for in the FLAKE model, but may strongly affect vertical mixing; (2) temperature of maximum density, which is different from that of fresh water and may affect winter convection; and (3) freezing temperature, which is different from that of fresh water and may affect the onset and duration of the ice cover period. While the first of these mechanisms is difficult to be included in the model designed for freshwater lakes, the other two, probably, could be taken into account, if it is possible to replace the respective constants in the model (i.e., the freezing temperature and the maximum density temperature) by those appropriate for Qinghai Lake. I suspect that the exact values of either variable for the Qinghai are unknown because of the lack of direct measurements and because the ionic composition of the lake is different from that of the ocean. However, as a "first guess", the oceanic values for the respective salinity 12.5 g/L can be considered - namely, about -0.65oC for freezing point, and about 1.6oC for TMD. If it is possible to repeat some of the experiments done using FLAKE with the settings modified accordingly, and then assess the differences in the outcomes of the "freshwater" and "salty" experiments, this would allow to evaluate the role of salinity vs air temperature and surface fluxes and hence strengthen the study. If this approach is technically not possible, potential role of salinity still should be discussed in the paper, perhaps based on literature and data from other similar lakes.

More specific comments:

The "Study area" Section: The elevation of the Tibetan Plateau is never mentioned in the paper. What is the absolute elevation of Qinghai Lake surface? This is an important piece of information, please specify.

P3 Lines 25-30: It follows from these numbers that the lake's water budget has been shifting towards an increase of the incoming components since 1970, accompanied by the decrease of evaporation. Then why the lake kept shrinking until 2004? Was the rate of shrinking in the 1960s much higher than in the early 2000s? Please explain.

P4 L15: "rare abnormal values influenced probably by cloud cover" – if you are confident that these abnormal values are artifacts corresponding to low clouds, then why keep them? Just remove them from your data base and the plot.

P5 Section 2.2: More details about the FLAKE model would be useful. What is the form of the expression for the profile in the lower layer?

P5 L30 and thereafter: The adjustments introduced to the air temperature and wind speed through linear regressions seem to help very little in minimizing biases between the simulated and the observed LST, so what is the point of using them?

L11 P15: "Keeping in mind the cool skin effect, we can suggest that the model predictions of the bulk LST are even better than the satellite data suggest" – But your Figure 2 shows good agreement between the satellite and the buoy data, and the latter meadured bulk temperature. Therefore, it looks like the skin effect in this case did not affect much the satellite-derived temperatures.

―――――――――――――――――――――――

---

## Referee Comment (RC2) · Anonymous Referee #2 · 14 Jan 2019

The Qinghai lake is the largest in land lake in China. It has large volume of biotic resources and tourism resources. Its thermodynamic changes under global warming remains unclear. Su et al. use a one-dimensional lake model to investigate thermodynamic changes of the Qinghai lake in the last three decades. The results show that the Qinghai lake has been warming up in the last three decades and the warming was the strongest in winter. Before getting published, however, this manuscript should be revised in several aspects. Please consider the points listed below and marked out in the manuscript. I strongly recommend language editing by some native English speaker, there are many errors in the grammar and improper expressions. 1. The authors emphasize the ice cover plays the first role in long-term change of thermodynamics, however, they do not validate the performance of Flake on the ice dynamics.

The ice-on and ice-off dates can be obtained from MODIS data. The authors can use the MODIS-derived ice-on and -off dates to validate the performance of the Flake on ice phenology. Here is the data link: http://www.csdata.org/p/214/ 2. The author should deemphasize the purpose to validate the performance of Flake on the Tibetan Plateau. Because both Lazhu et al. (2016) and Kirillin et al. (2017) has demonstrated its performance on the Tibetan Plateau. in Lazhu's study, the Namco lake is also a brackish and large lake. they even use observational temperature at different depths to validate its performance. in this respect, their study should be a better case to evaluate the performance of the Flake Model. 3. Even the Qinghai lake is a brackish lake, but its salinity is not low ($\sim$12.5 g/L). I agree with the authors that the salinity would not change the mixing type (dimictic), but salinity produces effects on the dates of spring and autumn overturning, which will change the energy flux. do the authors have any other observation data related the mixing of lake water column? If they have, they should show it or have some description on it.

I have some other comments and suggestions, please find the attached PDF file for details.

Please also note the supplement to this comment:
https://www.hydrol-earth-syst-sci-discuss.net/hess-2018-583/hess-2018-583-RC2-supplement.pdf

**Supplement:**

[revised manuscript text omitted]

---

## Author Comment (AC1) · 7 Mar 2019

**Responses to the comments from the Referee #1**

We are very grateful to the Referee for the comments on our manuscript. Those comments are all valuable and very helpful for revising and improving our manuscript. We have substantially revised our manuscript after reading the comments. The Referee's comments are shown in bold and our responses immediately follow.

**The paper is focused on Qinghai Lake, the largest of thousands of lakes situated on Tibetan Plateau, China. The lake is brackish with salinity about 12.5 g/L. The authors use the well-known one-dimensional model FLAKE forced with a local set of historical gridded meteorological data for the period 1979-2012 to simulate the thermal and ice regimes and their ongoing trends accompanying the global warming. Because the Qinghai, as well as all other Tibetan lakes, has been very sparsely covered by in situ measurements, and virtually no field monitoring data are available (except those from a single meteorological buoy used in this study), numerical simulation is the only mean capable of giving quantitative insights into the long-term variability of the Tibetan lakes. Therefore, in my opinion, the article presents interesting and useful information and should be published after moderate revision.**

**Comment #1:**

**My only general comment about this manuscript is as follows: I think that the possible role of salinity and its changes in the estimated long-term variability of the lake regime should be evaluated and discussed more thoroughly. For instance, can the trends of the ice regime (section 3.3) be associated not only with the air temperature increase, but also, at least partly, with salinity increase over the period 1961-2004? According to the information supplied in section 2.1, the lake level dropped for about 3.3 m during this period, which, given the mean depth of 21 m and mean salinity about 10 g/L, implies salinity increase of about 2 g/L. This, in turn, may have affected the ice regime.**

**Generally, salinity may exercise influence on the issues addressed in the article through (1) salinity stratification, which is not accounted for in the FLAKE model, but may strongly affect vertical mixing; (2) temperature of maximum density, which is different from that of fresh water and may affect winter convection; and (3) freezing temperature, which is different from that of fresh water and may affect the onset and duration of the ice cover period. While the first of these mechanisms is difficult to be included in the model designed for freshwater lakes, the other two, probably, could be taken into account, if it is possible to replace the respective constants in the model (i.e., the freezing temperature and the maximum density temperature) by those appropriate for Qinghai Lake. I suspect that the exact values of either variable for the Qinghai are unknown because of the lack of direct measurements and because the ionic composition of the lake is different from that of the ocean. However, as a "first guess", the oceanic values for the respective salinity 12.5 g/L can be considered - namely, about -0.65 °C for freezing point, and about 1.6 °C for TMD. If it is possible to repeat some of the experiments done using FLAKE with the settings modified accordingly, and then assess the differences in the outcomes of the "freshwater" and "salty" experiments, this would allow to evaluate the role of salinity vs air temperature and surface fluxes and hence strengthen the study. If this approach is technically not possible, potential role of salinity still should be discussed in the paper, perhaps based on literature and data from other similar lakes.**

**Author's response:** Thanks for the good evaluation and kind suggestions. We agree with the Reviewer that the salinity effects deserve an extended discussion. The influence of salinity changes over the period 1961-2004 on ice regime trend can indeed be hypothesized. However, quantification of the salinity effect and its comparison with the air temperature influence needs a separate in-depth study. First, specific changes in the salinity of Qinghai Lake require a stronger data support than an approximate estimation from lake depth changes. However, at present, we do not have historical data on the change of salinity in Qinghai Lake. Second, apart from air temperature and salinity, the ice regime is also influenced by other factors, such as wind, water circulation under ice cover and precipitation, which should be taken in to account, but are not considered in the framework of 1-D modeling. Hence, the effect of changes in salinity on lake ice regime cannot be clearly distinguished from other factors by using FLake model with a simple salinity parameterization. Last, the simulated ice durations were shortened 21.1 days and 26.6 days during 26 years from 1979 to 2004 for the freshwater and salt water simulations respectively. After considering the 12.5 g/l salinity effects on temperature of maximum density and freezing point, the average ice duration reduced ~13.8 days (Fig. 7 in revised manuscript, already enclosed). When the salinity changes ~1.2 g/l from 1979 to 2004 (correspond to 2g/l from 1961 to 2004), the ice duration may approximately reduce ~1.3 days, which is much litter than 26.6 days caused by meteorological factors. The influence of lake level caused salinity change on ice duration can be ignored compare to meteorological factors here. Hence, we considered the salinity effects but ignored its variation in the study, and focused on the lake response to the meteorological forcing. In reply to this useful comment, we added these considerations to the discussion (see the changes in the manuscript below) and will consider it in future research.

Just like what the referee said and we mentioned above, the mechanism of salinity stratification is difficult to be included in the model designed for freshwater lakes and the vertical salinity gradient of the lake was scarcely observed. We agreed with the Reviewer's suggestion and parameterized the salinity effects on the temperature of maximum density ($T_m$) and freezing point ($T_f$) into the lake model based on linear approximations of empirical function of state of seawater, then rerun the Flake model. With the consideration of salinity effects, the lake ice phenology had been improved. Correspondingly, the simulation results were changed respectively, and the relevant parts of the manuscript were revised. The major quantitative conclusions of the study remained unchanged.

The mainly relevant parts in the manuscript were also revised as follows:
(1) In section 2.3 Lake model, the parameterizations of salinity effects were added:
"Considering that we are applying a freshwater lake model to a brackish lake, the equation of state used by FLake was adjusted by changing temperature of maximum water density ($T_m$) and freezing temperature ($T_f$). The $T_m$ and $T_f$ formula obtained from linear approximations of empirical function of state of seawater (Caldwell, 1978; UNESCO, 1981) are:

$$T_m[°C] = 3.98-0.216S \qquad (5)$$

$$T_f[°C] = -0.055S \qquad (6)$$

Where the S is salinity taken in parts per thousand (‰ or g·L$^{-1}$). For the salinity of S=12.5g/L, which is the case of Qinghai Lake, the equation gives $T_m$ = 1.28°C and $T_f$= -0.69°C."

(2)  In section 3.3 Lake ice cover, the following are added:

"Salinity had a certain effect on the lake ice regime after parameterizations on temperature of maximum density and freezing point (Fig. 7 in revised manuscript, enclosed). The maximum ice thickness was reduced compared with default (freshwater) model setting. The freeze-up data was delayed and the break-up data was advanced, leading to a shorter ice duration period. Nevertheless, the interannual changes between them remained consistent and the break-up date and ice duration period are closer to observation."

(3)  In part 4.1 Model performance, the following are added:

"Salinity can influence the temperature of maximum density ($T_m$) and the freezing temperature of water ($T_f$). According to the 12.5g/L salinity of Qinghai Lake, these two parameters equal to 1.28 °C and -0.69 °C instead of the default model configurations of 4 °C and 0 °C respectively. Considerations of the salinity effects lead to a slightly earlier of the spring overturn and a later autumn overturn and consequently to an extension of the lake stratification period. Because the salinity stratification effects cannot be completely included in the model designed for freshwater lakes, the simulated mixing regime may have some differences from the actual situation of Qinghai Lake. The major features of the dimictic seasonal mixing regime, as simulated by the model, may be suggested to be close to the reality."

(4)  And the following are also added in part 4.1:

"Despite a salinity parameterization, the ice phenology modelled by FLake differs significantly from the remote sensing observations. The discrepancy may be related to a number of factors not included in the model. One of them is the effect of salinity on the ice structure, density, and porosity; the others are precipitation, inflows, circulation under ice cover and wind, which is especially important of large-area lakes, like Qinghai Lake (Kirillin et al. 2012). However, the air temperature apparently has the strongest effect on ice regime, especially in long-term changes, which appear to be well-simulated by FLake allowing us to study the effect of air temperature on lake ice regime within the model ability."

[Figure]

**Figure 7: The interannual variations of simulated maximum annual ice thickness (a), break-up date (b), freeze-up date (c) and ice duration period (d) of Qinghai Lake. The coloured (grey) line indicate with (without) salinity parameterization. The red dash line is air temperature anomaly in the specified period and the black line is observed ice phenology derived from satellite.**

**More specific comments:**

**Comment #1**

**The "Study area" Section: The elevation of the Tibetan Plateau is never mentioned in the paper. What is the absolute elevation of Qinghai Lake surface? This is an important piece of information, please specify.**

**Author's response:** The elevation of the Qinghai Lake surface already given in brackets of the first sentence in "study area" section. Maybe it is not very obvious in brackets, so I replaced it with a sentence to specify the elevation of Qinghai Lake. In the revised manuscript, the sentence in "study area" section was changed to:

"It is an endorheic, brackish lake (salinity 12.5 g·L$^{-1}$, pH 9.3) (Deng et al., 2010) located on the northeast margin of the TP (Fig. 1) at the height of about 3194 m. a.s.l."

**Comment #2**

**P3 Lines 25-30: It follows from these numbers that the lake's water budget has been shifting towards an increase of the incoming components since 1970, accompanied by the decrease of evaporation. Then why the lake kept shrinking until 2004? Was the rate of shrinking in the 1960s much higher than in the early 2000s? Please explain.**

**Author's response:** The change in lake water level depends on the balance between incoming components and evaporation loss. Many studies on the water level changes in Qinghai Lake show that the water level in Qinghai Lake was declining in fluctuations from 1959 to 2004 (e.g. Zhang et al.,2011; Li et al., 2007; Tang et al., 2018; all cited in manuscript). This is because the water loss (i.e. evaporation) is generally larger than incoming water (e.g. runoff and precipitation) in this period, although the incoming components is increasing and the evaporation is decreasing (Tang et al., 2018), it still need some time to get balance with the water loss, so the lake kept shrinking until 2004 when the increasing incoming water balanced with the water loss. The rate of shrinking of Qinghai Lake in the early of 1960s indeed seems much higher than in the early 2000s (Zhang et

al.,2011), but it turned to a temporary expandation in the late of 1960s (e.g. Zhang et al., 2011; Li et al., 2007). We have revised the last paragraph of section 2.1 in manuscript to explain the reason (Page 3 Line 26-32):

"Qinghai Lake is sensitive to climate change. Because the evaporation is generally larger than river runoff and precipitation, the water level of Qinghai Lake decreased at the average rate of 7.6 cm per year from 1961 to 2004 (Cui et al., 2016). However, the precipitation continuously increased in 1970-2015 by 15.603 mm per decade according to the data from Gangcha station (the nearest meteorological station approximately 13 km north to Qinghai Lake), and the runoff from the melting of Qilian Mountain glaciers is also increasing because of an increasing temperature of 0.3 °C per decade from 1961 to 2012 in Qinghai lake basin, coupled with the decreasing evaporation by 1.343 mm per year (observed by Gangcha station) during 1970-2003 (Tang et al., 2018). Since 2004, as the runoff and precipitation begin to exceed evaporation, the Qinghai lake stopped shrinking and began to expand, lake level increased at a rate of 14 cm per year during the period 2004-2012. The regional climate gradually turned to the direction of "warm and humid" (Dong and Song, 2011; Zhang et al., 2011, 2014b; Cui et al., 2016)."

**Comment #3**

**P4 L15: "rare abnormal values influenced probably by cloud cover" – if you are confident that these abnormal values are artifacts corresponding to low clouds, then why keep them? Just remove them from your data base and the plot.**

**Author's response:** We have removed the abnormal values according to the referee's comment. And the relevant sentence in the manuscript are changed as follows:

"We have removed few abnormal values that maybe influenced by cloud cover (Langer et al., 2010)."

**Comment #4**

**P5 Section 2.2: More details about the FLake model would be useful. What is the form of the expression for the profile in the lower layer?**

**Author's response:** We are very sorry for the negligence of details about FLake. According to Referee's comment, we added the expression in FLake for the temperature profile in the lower layer in the "2.3 lake model" section of revised manuscript as follows:

"The parameterization formula is:

$$\frac{\theta_s(t) - \theta(z,t)}{\Delta\theta(t)} = \Phi_\theta(\zeta) \quad h(t) \leq z \leq D \tag{1}$$

Where $t$ is time, $z$ is the depth, $\theta_s(t)$ is the temperature of upper mixed layer of depth $h(t)$, which is computed based on the convective entrainment or relaxation-type equation in terms of wind mixing (Mironov, 2008). $\Delta\theta(t) = \theta_s(t) - \theta_b(t)$ is the temperature differences across the thermocline with the depth of $\Delta h(t) = D - h(t)$, $D$ is the lake depth, $\theta_b(t)$ is the temperature at the bottom of the thermocline. $\Phi_\theta(\zeta)$ is a dimensionless "universal" function of the dimensionless depth $\zeta = \frac{z - h(t)}{\Delta h(t)}$ which satisfies the boundary conditions $\Phi_\theta(0) = 0$ and $\Phi_\theta(1) = 1$. Based on the above theory, the temperature profile parameterization of the two-layer at time $t$ can be obtained:

$$\theta(t) = \begin{cases} \theta_s(t) & 0 \le z \le h(t) \\ \theta_s(t) - [\theta_s(t) - \theta_b(t))\Phi_\theta(\zeta)] & h(t) \le z \le D \end{cases} \qquad (2)$$

The shape function $\Phi_\theta(\zeta)$ is calculated by:

$$\Phi_\theta = \left(\frac{40}{3}C_\theta - \frac{20}{3}\right)\zeta + (18 - 30C_\theta)\zeta^2 + (20C_\theta - 12)\zeta^3 + (\frac{5}{3} - \frac{10}{3}C_\theta)\zeta^4 \qquad (3)$$

The shape factor $C_\theta$ can be computed by:

$$\frac{dC_\theta}{dt} = sign\left(\frac{dh(t)}{dt}\right)\frac{C_\theta^{max} - C_\theta^{min}}{t_{rc}} \qquad C_\theta^{min} \le C_\theta \le C_\theta^{max} \qquad (4)$$

Where $t_{rc}$ is the relaxation time scale (s) which is the time of the evolution of the temperature profile in the thermocline from one limiting curve to the other, and sign is the signum function, following the change of sign in $\frac{dh(t)}{dt}$. $C_\theta^{min} = 0.5$ and $C_\theta^{max} = 0.8$ are the minimum and maximum values of the shape factor."

**Comment #5**

**P5 L30 and thereafter: The adjustments introduced to the air temperature and wind speed through linear regressions seem to help very little in minimizing biases between the simulated and the observed LST, so what is the point of using them?**

**Author's response:** The buoy observation data available for bias correction were unfortunately not complete, covering only summer and autumn and some of the data are missing. No correction was performed for other parts of the year; therefore an appreciable bias remained in the results. In addition, the revised air temperature and wind speed may not have enough consistency to use, but it helps in understanding and evaluating the bias caused by the forcing data. We revised the sentence in section 3.1:

"Through the correction of the driving data, we found that positive bias between simulated LST and satellite data can be partly explained by the differences in the forcing weather data measured over the lake and provided by the ITPCAS data, while the remaining bias may be partly attributed to the cool skin effect in the LST sensed by MODIS."

**Comment #6**

**L11 P15: "Keeping in mind the cool skin effect, we can suggest that the model predictions of the bulk LST are even better than the satellite data suggest" – But your Figure 2 shows good agreement between the satellite and the buoy data, and the latter measured bulk temperature. Therefore, it looks like the skin effect in this case did not affect much the satellite-derived temperatures.**

**Author's response:** The satellite observed LST have a little negative bias (-0.36 °C, in summer and autumn) compared with the buoy bulk temperature. While the y-scale of Fig. 2 do not allow to see the bias clearly, the bias values are added to the panels on the figure. If the skin effect, in this case, did not affect much the satellite-derived temperatures, it probably caused by the different measurement methods between satellite and buoy. We have rewritten this part according to the referee's comment and removed the sentence:

"Keeping in mind the cool skin effect," and add the sentence "Meanwhile, the difference of the measurement method between satellite and buoy may also play a part in it (Leppäranta and Lewis, 2007)."

**Other changes:**

We tried our best to improve the manuscript and made some changes in the manuscript. These changes will not influence the content and framework of the paper. Many grammatical or typographical errors have been revised but not list here. The manuscript will be further revised by the English editorial company.

**References:**

Caldwell, D., R.: The maximum density points of pure and saline water, Deep Sea Res.,25, 175–181, doi: 10.1016/0146-6291(78)90005-x,1978.

Cui, B. L. and Li, X. Y.: The impact of climate changes on water level of Qinghai Lake in China over the past 50 years, Hydrol. Res., 47, 532–542, doi:10.2166/nh.2015.237, 2016.

Deng, S., Dong, H., Lv, G., Jiang, H., Yu, B., and Bishop, M. E.: Microbial dolomite precipitation using sulfate reducing and halophilic bacteria: Results from Qinghai Lake, Tibetan Plateau, NW China, Chem. Geol., 278, 151–159, doi:10.1016/j.chemgeo.2010.09.008, 2010.

Dong, H. and Song, Y.: Shrinkage history of Lake Qinghai and causes during the last 52 years, 2011 International Symposium on Water Resource and Environmental Protection (ISWREP), Xi'an, China, 20-22 May 2011, 446–449, 2011.

Kirillin, G., Leppäranta, M., Terzhevik, A., Granin, N.,Bernhardt, J., Engelhardt, C,. Efremova, T., Golosov, S., Palshin, N., Sherstyankin, P., Zdorovennova, G., and Zdorovennov, R.: Physics of seasonally ice-covered lakes: a review, Aquat. Sci., 2012, 74, 659-682, 2012.

Langer, M., Westermann, S., Boike, J.: Spatial and temporal variations of summer surface temperatures of wet polygonal tundra in Siberia-implications for MODIS LST based permafrost monitoring, Remote Sens. Environ., 114, 2059–2069, doi:10.1016/j.rse.2010.04.012, 2010.

Leppäranta, M., Lewis, J. E.: Observations of ice surface temperature and thickness in the Baltic Sea, Int. J. Remote Sens., 28, 3963–3977, doi:10.1080/01431160601075616, 2007.

Tang, L., Duan, X., Kong, F., Zhang, F., Zheng, Y., Li, Z., Mei, Y., Zhao, Y., and Hu, S.: Influences of climate change on area variation of Qinghai Lake on Qinghai-Tibetan Plateau since 1980s, Sci. Rep., 8, 7331, doi:10.1038/s41598-018-25683-3, 2018.

UNESCO, The Practical Salinity Scale 1978 and the International Equation of State of Seawater 1980. Technical Paper in Marine Science, 36,1981.

Zhang, G., Xie, H., Kang, S., Yi, D., and Ackley, S. F.: Monitoring lake level changes on the Tibetan Plateau using ICESat altimetry data (2003–2009), Remote. Sens. Environ., 115, 1733–1742, doi:10.1016/j.rse.2011.03.005, 2011.

Zhang, G., Yao, T., Xie, H., Zhang, K., and Zhu, F.: Lakes' state and abundance across the Tibetan Plateau, Sci. Bull. (Beijing), 59, 3010–3021, doi:10.1007/s11434-014-0258-x, 2014b.

---

## Author Comment (AC2) · 7 Mar 2019

**Responses to the comments from the Referee #2**

We are very grateful to the Referee for the comments on our manuscript. Those comments are all valuable and very helpful for revising and improving our manuscript. We have substantially revised our manuscript after reading the comments. The Referee's comments are shown in bold and our responses immediately follow.

**The Qinghai lake is the largest in land lake in China. It has large volume of biotic resources and tourism resources. Its thermodynamic changes under global warming remains unclear. Su et al. use a one-dimensional lake model to investigate thermodynamic changes of the Qinghai lake in the last three decades. The results show that the Qinghai lake has been warming up in the last three decades and the warming was the strongest in winter. Before getting published, however, this manuscript should be revised in several aspects. Please consider the points listed below and marked out in the manuscript. I strongly recommend language editing by some native English speaker, there are many errors in the grammar and improper expressions.**

**Author's response:** Many grammatical or typographical errors have been revised but not list here. The manuscript will be further revised by the English editorial company.

**Comment #1**

**The authors emphasize the ice cover plays the first role in long-term change of thermodynamics, however, they do not validate the performance of Flake on the ice dynamics. The ice-on and ice-off dates can be obtained from MODIS data. The authors can use the MODIS-derived ice-on and -off dates to validate the performance of the Flake on ice phenology. Here is the data link: http://www.csdata.org/p/214/.**

**Author's response:** Thanks for the referee's suggestion. As referee's suggestion, we use the MODIS-derived ice-on and ice-off dates validate the Flake on ice phenology. In the revised manuscript, we add the MODIS-derived ice phenology in Fig.7 (in revised manuscript, enclosed), and add the description in section 3.3 Lake ice cover:

"Salinity had a certain effect on the lake ice regime after parameterizations on temperature of maximum density and freezing point (Fig. 7). The maximum ice thickness was reduced compared with default (freshwater) model setting. The freeze-up data was delayed and the break-up data was advanced, leading to a shorter ice duration period. Nevertheless, the interannual changes between them remained consistent and the break-up date and ice duration period are closer to observation."

[Figure]

**Figure 7: The interannual variations of simulated maximum annual ice thickness (a), break-up date (b), freeze-up date (c) and ice duration period (d) of Qinghai Lake. The coloured (grey) line indicate with (without) salinity parameterization. The red dash line is air temperature anomaly in the specified period and the black line is observed ice phenology derived from satellite.**

**Comment #2**

**The author should deemphasize the purpose to validate the performance of FLake on the Tibetan Plateau. Because both Lazhu et al. (2016) and Kirillin et al. (2017) has demonstrated its performance on the Tibetan Plateau. In Lazhu's study, the Nam Co lake is also a brackish and large lake. They even use observational temperature at different depths to validate its performance. In this respect, their study should be a better case to evaluate the performance of the Flake Model.**

**Author's response:** Thanks for the Referee's kind advice. We agree with the reviewer: the studies of Lazhu et al (2016) and Kirillin et al (2017) considered several aspects of the performance of FLake on the Tibetan Plateau lakes. The first study was focused on the evaporation estimations at Nam Co Lake (salinity ~1.78 g l$^{-1}$), the second one considered thermal regime of freshwater lakes. The present study comprehensively tests the FLake performance on the largest, brackish lake of the Plateau as a "worst-case" test for a 1-D freshwater model. It suggests that the results are extendable on the vast majority of the Tibetan lake system with at least the same or better performance. And in the revised manuscript, the salinity effects on temperature of maximum density and freezing point had been considered based on the comments, and the model performance has been improved. According to the reviewer's suggestion, we deemphasize the purpose to evaluate Flake but still mentioned one of our study purposes was to evaluate the FLake model in the Tibetan Plateau. The changed content in the revised manuscript in section 4.1 Model performance were as follows:

"Following the study of Lazhu et al. (2016) at Nam Co Lake (salinity ~1.78 g l$^{-1}$) and Kirillin et al. (2017) at Ngoring Lake (a freshwater lake), the good prediction of the LST over the largest, brackish lake of Tibet Tibetan by the relatively simple, highly parameterized model FLake, verified by satellite and buoy data, is one of the core results of this study."

**Comment #3**

**Even the Qinghai lake is a brackish lake, but its salinity is not low (~12.5 g/L). I agree with the authors that the salinity would not change the mixing type (dimictic), but salinity produces effects on the dates of spring and autumn overturning, which will change the energy flux. do the authors have any other observation data related the mixing of lake water column? If they have, they should show it or have some description on it.**

**Author's response (see also the reply to the first comment of Reviewer#1):**

Thanks for the referee's good evaluation and kind suggestion. In order to additionally investigate the potential salinity effects, we rerun the FLake model coupled with the salinity effects on temperature of maximum density and freezing point, and found that the salinity had an effect on the dates of spring and autumn overturning. The energy fluxes were also reanalyzed based on the new simulation results that were improved but did not change the main conclusions. More observation data on mixing of lake water column are required for a more thorough analysis of salinity influence on stratification. As described above in response to Referee #1, We added the following text to Section 4.1 "Model performance":

"Salinity can influence the temperature of maximum density ($T_m$) and the freezing temperature of water ($T_f$). According to the 12.5g/L salinity of Qinghai Lake, these two parameters equal to 1.28 °C and -0.69 °C instead of the default model configurations of 4 °C and 0 °C respectively. Considerations of the salinity effects leads to a slightly earlier of the spring overturn and a later autumn overturn and consequently to an extension of the lake stratification period. Because the salinity stratification effects cannot be completely included in the model designed for freshwater lakes, the simulated mixing regime may have some differences from the actual situation of Qinghai Lake. The major features of the dimictic seasonal mixing regime, as simulated by the model, may be suggested to be close to the reality."

**I have some other comments and suggestions, please find the attached PDF file for details.**

**Comments in the supplement:**

**Comment #1**

**P3 L20: "The lake usually freezes up in December/January and the ice breaks up in early April." Where dose these data come from? Your observation or reference.**

**Author's response:** These data are come from the published paper by Li et al (2016). The reference was already included in the paragraph. We are very sorry for our unclear expression. To clarify the data source, we changed the corresponding part as follows:

"The lake is ice-covered from December/January to early April; the average annual lake water temperature is 5.4 °C, with the maximum monthly temperature of 17.2 °C (August) and the minimum of -2.0 °C (January) (Li et al., 2016)."

**Comment #2**

**P4 L5: Do you have water temperature observations on different depths? If you have, you'd better show it.**

**Author's response:** No, we don't have the observations on different depths so far.

**Comment #3**

**P5 L5: What is the version of the Flake model?**

**Author's response:** We download the official version of FLake from the model's website http://www.flake.igb-berlin.de/index.shtml, and have already described it in section Data availability in the revised manuscript as:

"The lake model FLake is available from the model community site (http://www.flake.igb-berlin.de/index.shtml)."

**Comment #4**

**P5 L12: You do not have to emphasize the computational efficiency to much which has been emphasized in row-5, page-3. Considering the computing ability of current PCs, almost all one-dimensional or two-dimensional models can be well executed.**

**Author's response:** According to the comment, we deleted the sentence below that emphasize the computational efficiency in the revised manuscript:

"This simple two-layer parameterization of the water column provides FLake with computational efficiency, while preserves the essential physics."

**Comment #5**

**P7 L5: what does this mean? do you mean the break-up and freeze-up dates are the most sensitive proxy to all the meteorological parameters, including air temperature, solar radiation, and wind?**

**Author's response:** We removed the sentence below in revised manuscript:

"The variations of break-up and freeze-up dates are sensitive to the meteorological conditions, in the first place, air temperature, solar radiation, and wind."

**Comment #6**

**P7 L16: What is the direction of the energy flux? Outgoing-negative, incoming-positive?**

**Author's response:** For radiation flux and net energy flux, downward is positive and upward is negative. For SH, LH and lake surface released heat flux, upward is positive and downward is negative.

**Comment #7**

**P8 L2: I am confused by the logic of this part. based on the first sentence, I thought that the authors wanted to discuss how the lake-air temperature difference determined the energy flux. but after reading the following two paragraphs, I realized that the authors discussed how the different fractions of the energy flux contributed to the lake-air temperature difference.**

**Author's response:** The first two sentences of this paragraph are contradictory, we deleted the first sentence:

"The lake-air temperature difference is one of the important factors determining the surface heat exchange." in revised manuscript.

**Comment #8**

**P11: I was confused by this paragraph. as the Figure 6 shows, not only air temperature and wind speed, but also long-wave radiation and short-wave radiation have obvious correlation**

with water temperature. Long-wave radiations even have the highest correlation coefficient. Why the author only claims changes in windspeed and air temperature drive the lake warming? Furthermore, the net long-wave radiation has a trend of -1.86 W/m²/decade, the net short-wave radiation has a trend of 2.35 W/m²/decade based on the Figure 8. These trends are much larger the trends of sensible heat flux and latent heat flux which are functions of lake-air temperature difference and wind speed. Please consider the roles of short- and long-radiation in lake warming, and add corresponding stuff in your conclusion.

**Author's response:** The air temperature and longwave radiation are obviously correlated, because the downward longwave radiation is function of the air temperature in the fourth degree (affected by emissivity properties of the atmosphere). Hence, the air temperature is the major factor here. The shortwave radiation is negatively and insignificantly (R<0.3) correlated with the water temperature, and do not contribute to the water temperature increase (although can explain the slower increase of the water temperature compared with the air temperature [see Kirillin et al. 2017]). Hence, air temperature remains to be the major factor affecting the long-term trend in water temperatures.

According to the comment, we revised the sentences in section 4.2 as follows:

"As expected, the correlation analysis shows that the changes in LST were closely related to air temperatures, downward long-wave radiation and wind speed (Fig. 6). The increase of the air temperature and downward long-wave radiation plays a key role in lake surface temperature warming, and the decrease in wind speed also promoted the warming of the lake surface temperature. The shortwave radiation is negatively and insignificantly (R<0.3) correlated with the water temperature, and do not contribute to the water temperature increase (although can explain the slower increase of the water temperature compared with the air temperature [see Kirillin et al. 2017]). The decrease in ice cover duration increases in turn the annual amount of short-wave radiation penetrating to the water column, accelerating the warming."

**Comment #9**
**P12: For low-altitude lakes in the mid-latitudes? you'd better list at least one reference paper here.**

**Author's response:** According to the comment, we add several references in the revised manuscript and reorganized the language in section 4.3 as follows:

"For low-altitude temperate and boreal lakes, the air temperatures are typically higher than LST after the ice-off and remain higher until temperature equilibrate around mid-summer. In the subsequent period down to ice-on, the LST is typically higher than the air temperature. Hence, the atmospheric boundary layer is generally stable throughout much of the summer season over low-altitude lakes (Rouse et al., 2003; Nordbo et al., 2011; Momii and Ito, 2008; Gianniou and Antonopoulos, 2007; Scott and Huff, 1996)."

**Comment #10**
**P26 Fig. 7: Do you have observational data of ice thickness?  ~1.1 m is very thick for lake ice. The thickest ice on the Namco lake is 70 cm based on the observation (Qu et al., 2012, Chinese with English abstract, Lake Ice and Its Effect Factors in the Nam Co Basin, Tibetan Plateau) where is 1500 m higher than the Qinghai lake.**

**Author's response:** We do not have observational data of ice thickness. The simulated maximum ice thickness of ~1.1 m was the situation in 1980s, in the later 2000s, the thickness of the lake ice

had dropped significantly to about 0.7 m, same with Namco in this period according to Qu et al (2012). And after considering the salinity effect, the simulated maximum ice thickness further reduced (Fig. 7 in revised manuscript, enclosed). In addition, the Namco Lake (33 m for average depth) is deeper than Qinghai Lake (21 m for average depth), which also may result in different lake ice thicknesses.

**Comment #11**

**P12: Do not understand what the blue bars and red bars represent respectively in Figure 9a**

**Author's response:** According to the comment, we add the explanation to the legend of figure 9 as follows:

"Figure 9: Climatological mean seasonal variations (5-day moving average, lines) in simulated LST and air temperature (a) with their difference (b), downward shortwave radiation(c), downward longwave radiation(d), net shortwave radiation (e) and net longwave radiation (f). The bars indicate monthly averaged mean annual variation trend (red for positive and blue for negative except in (a) that for air temperature and LST respectively) from 1979 to 2012. Solid points at end of the bars mean pass significance test of $p<0.01$ and hollow points mean $p<0.05$. The grey areas indicate the freeze-up and break-up date variation range of the lake."

**References:**

Gianniou, S. K., Antonopoulos, V. Z.: Evaporation and energy budget in Lake Vegoritis, Greece, J. of Hydrol., 345, 212-223, doi: 10.1016/j.jhydrol.2007.08.007, 2007.

Kirillin, G., Wen, L., and Shatwell, T.: Seasonal thermal regime and climatic trends in lakes of the Tibetan highlands, Hydrol. Earth Syst. Sci., 21, 1895–1909, doi:10.5194/hess-21-1895-2017, 2017.

Lazhu, Yang, K., Wang, J., Lei, Y., Chen, Y., Zhu, L., Ding, B., and Qin, J.: Quantifying evaporation and its decadal change for Lake Nam Co, central Tibetan Plateau, J. Geophys. Res. Atmos., 121, 7578-7591, doi:10.1002/2015JD024523, 2016.

Li, X. Y., Ma, Y. J., Huang, Y. M., Hu, X., Wu, X. C., Wang, P., Li, G. Y., Zhang, S. Y., Wu, H. W., Jiang, Z. Y., Cui, B. L., and Liu, L.: Evaporation and surface energy budget over the largest high‐altitude saline lake on the Qinghai‐Tibet Plateau, J. Geophys. Res. Atmos., 121, 10470-10485, doi: 10.1002/2016JD025027, 2016.

Momii, K., Ito, Y.: Heat budget estimates for lake Ikeda, Japan, J. Hydrol., 361, 3-4, 362-370, doi: 10.1016/j.jhydrol.2008.08.004, 2008.

Nordbo, A., Launiainen, S., Mammarella, I., Leppäranta, M., Huotari, J., Ojala, A., and Vesala, T.: Long-term energy flux measurements and energy balance over a small boreal lake using eddy covariance technique, J. Geophys. Res., 116, D02119, doi:10.1029/2010JD014542, 2011.

Qu, B., Kang, S., Chen, F., Zhang, Y., and Zhang, G.: Lake Ice and Its Effect Factors in the Nam Co Basin, Tibetan Plateau, Progressus Inquisitiones De Mutatione Climatis, 8, 327-333, 2012.

Rouse, W. R., Oswald, C. M., Binyamin, J., Blanken, P. D., Schertzer, W. M., and Spence, C.: Interannual and seasonal variability of the surface energy balance and temperature of central Great Slave Lake, J. Hydrometeorol., 4, 720–730, doi:10.1175/1525-7541(2003)004<0720:IASVOT>2.0.CO;2, 2003.

Scott, R. W., Huff, F. A.: Impacts of the Great Lakes on regional climate conditions, J. Gt. Lakes Res., 22, 845-863, doi:10.1016/S0380-1330(96)71006-7, 1996.

---

## Author Response (AR1)

Dear Miguel Potes,

On behalf of my co-authors, we thank you very much for giving us an opportunity to revise our manuscript, we appreciate editor and reviewers very much for their positive and constructive comments and suggestions on our manuscript entitled "Numerical study on the response of the largest lake in China to climate change". (MS No.: hess-2018-583).

We have tried our best to revise our manuscript according to the comments. Attached please find the revised version, which we would like to submit for your kind consideration.

We would like to express our great appreciation to you and reviewers for comments on our paper. Looking forward to hearing from you.

Thank you and best regards.

Sincerely yours,

Dongsheng Su

Corresponding author:

Name: Lijuan Wen

E-mail: wlj@lzb.ac.cn

We are very grateful to the Referee for the comments on our manuscript. Those comments are all valuable and very helpful for revising and improving our manuscript. We have substantially revised our manuscript after reading the comments. The Referee's comments are shown in bold and our responses immediately follow.

5    **Responses to the comments from the Referee #1**

**The paper is focused on Qinghai Lake, the largest of thousands of lakes situated on Tibetan Plateau, China. The lake is brackish with salinity about 12.5 g/L. The authors use the well-known one-dimensional model FLAKE forced with a local set of historical gridded meteorological data for the period 1979-2012 to simulate the thermal and ice regimes and**
10   **their ongoing trends accompanying the global warming. Because the Qinghai, as well as all other Tibetan lakes, has been very sparsely covered by in situ measurements, and virtually no field monitoring data are available (except those from a single meteorological buoy used in this study), numerical simulation is the only mean capable of giving quantitative insights into the long-term variability of the Tibetan lakes. Therefore, in my opinion, the article presents interesting and useful information and should be published after moderate revision.**

**Comment #1:**

**My only general comment about this manuscript is as follows: I think that the possible role of salinity and its changes in the estimated long-term variability of the lake regime should be evaluated and discussed more thoroughly. For instance, can the trends of the ice regime (section 3.3) be associated not only with the air temperature increase, but also,**
20   **at least partly, with salinity increase over the period 1961-2004? According to the information supplied in section 2.1, the lake level dropped for about 3.3 m during this period, which, given the mean depth of 21 m and mean salinity about 10 g/L, implies salinity increase of about 2 g/L. This, in turn, may have affected the ice regime.**
**Generally, salinity may exercise influence on the issues addressed in the article through (1) salinity stratification, which is not accounted for in the FLAKE model, but may strongly affect vertical mixing; (2) temperature of maximum density,**
25   **which is different from that of fresh water and may affect winter convection; and (3) freezing temperature, which is different from that of fresh water and may affect the onset and duration of the ice cover period. While the first of these mechanisms is difficult to be included in the model designed for freshwater lakes, the other two, probably, could be taken into account, if it is possible to replace the respective constants in the model (i.e., the freezing temperature and the maximum density temperature) by those appropriate for Qinghai Lake. I suspect that the exact values of either**
30   **variable for the Qinghai are unknown because of the lack of direct measurements and because the ionic composition of the lake is different from that of the ocean. However, as a "first guess", the oceanic values for the respective salinity 12.5 g/L can be considered - namely, about -0.65 °C for freezing point, and about 1.6 °C for TMD. If it is possible to repeat some of the experiments done using FLAKE with the settings modified accordingly, and then assess the differences in the outcomes of the "freshwater" and "salty" experiments, this would allow to evaluate the role of salinity**

**vs air temperature and surface fluxes and hence strengthen the study. If this approach is technically not possible, potential role of salinity still should be discussed in the paper, perhaps based on literature and data from other similar lakes.**

**Author's response:** Thanks for the good evaluation and kind suggestions. We agree with the Reviewer that the salinity effects deserve an extended discussion. The influence of salinity changes over the period 1961-2004 on ice regime trend can indeed be hypothesized. However, quantification of the salinity effect and its comparison with the air temperature influence needs a separate in-depth study. First, specific changes in the salinity of Qinghai Lake require a stronger data support than an approximate estimation from lake depth changes. However, at present, we do not have historical data on the change of salinity in Qinghai Lake. Second, apart from air temperature and salinity, the ice regime is also influenced by other factors, such as wind, water circulation under ice cover and precipitation, which should be taken in to account, but are not considered in the framework of 1-D modeling. Hence, the effect of changes in salinity on lake ice regime cannot be clearly distinguished from other factors by using FLake model with a simple salinity parameterization. Last, the simulated ice durations were shortened 21.1 days and 26.6 days during 26 years from 1979 to 2004 for the freshwater and salt water simulations respectively. After considering the 12.5 g/l salinity effects on temperature of maximum density and freezing point, the average ice duration reduced ~13.8 days (Fig. 7 in revised manuscript, enclosed). When the salinity changes ~1.2 g/l from 1979 to 2004 (correspond to ~ 2 g/l from 1961 to 2004), the ice duration may approximately reduce ~1.3 days, which is much smaller than 26.6 days caused by meteorological factors. The influence of lake level caused salinity change on ice duration can be ignored compare to meteorological factors here. Hence, we considered the salinity effects but ignored its variation in the study, and focused on the lake response to the meteorological forcing. In reply to this useful comment, we added these considerations to the discussion section in the revised manuscript (see changes in the manuscript below) and will consider it in future research.

Just like what the referee said and we mentioned above, the mechanism of salinity stratification is difficult to be included in the model designed for freshwater lakes and the vertical salinity gradient of the lake was scarcely observed. We agreed with the Reviewer's suggestion and parameterized the salinity effects on the temperature of maximum density ($T_m$) and freezing point ($T_f$) into the lake model based on linear approximations of empirical function of state of seawater, then rerun the Flake model. With the consideration of salinity effects, the lake ice phenology had been improved. Correspondingly, the simulation results were changed respectively, and the relevant parts of the manuscript were revised. The major quantitative conclusions of the study remained unchanged.

**Relevant changes made in the manuscript:**

(1) Page 5 Line 9, in section 2.3 Lake model, "To partially account for salinity effects in a brackish lake, the freshwater equation of state used by FLake was adjusted by changing temperature of maximum water density ($T_m$) and the freezing point temperature ($T_f$). The parameterization formula of $T_m$ and $T_f$ obtained from linear approximations of empirical function of state of seawater (Caldwell, 1978; UNESCO, 1981) are:

$$T_m[°C] = 3.98 - 0.216S \qquad (4)$$

$$T_f[°C] = -0.055S \qquad (5)$$

Where the S is salinity taken in parts per thousand (‰ or g l$^{-1}$). For the salinity of S=12.5 g l$^{-1}$, which is the case of Qinghai Lake, the equation gives $T_m$ = 1.28 °C and $T_f$= -0.69 °C." was added.

And in same section (P5 L20), "The model runs were performed using both original freshwater equation of state and the brackish water approximation (eq. 4-5). Here we defined the simulation with original freshwater equation of state as freshwater lake (FL) experiment and the simulation with the brackish water approximation as saltwater lake (SL) experiment." was also added.

(2) Page 7 Line 5, in section 3.3 Lake ice cover, "Compared with FL experiment, the salinity parameterization for $T_m$ and $T_f$ in SL experiment has a certain effect on the ice phenology (Fig. 7): the maximum ice thickness is reduced, the freeze-up date is delayed and the break-up date is advanced, leading to a shorter ice duration period. Nevertheless, the interannual changes between them remained consistent. The simulated freeze-up and break-up date in FL and SL experiments are both later than satellite observations, with some differences in interannual variations but similar range in ice duration. In SL experiment, the maximum ice thickness and the break-up date are closer to the observations, the former was reported of 0.7 m by Chen et al (1995). Hence the ice phenology results from SL experiment were used for further analysis." was added.

(3) Page 10 Line 16, in section 4.1 Model performance, the statements of "Still, since the model used here is the freshwater lake model, the simulated mixing regime may have some difference with the actual situation of Qinghai Lake." were corrected as: "Salinity can influence the temperature of maximum density ($T_m$) and the freezing temperature of water ($T_f$). According to the 12.5g l$^{-1}$ salinity of Qinghai Lake, these two parameters equal to 1.28 °C and -0.69 °C instead of the default model configurations of 4 °C and 0 °C respectively. Considerations of the salinity effects lead to a slightly earlier spring overturn and a later autumn overturn, and consequently to an extension of the lake stratification period. Because the salinity stratification effects cannot be completely included in the model designed for freshwater lakes, the simulated mixing regime may have some differences from the actual situation of Qinghai Lake."

(4) Page 10 Line 17, in section 4.1 Model performance, "Despite incorporation of the salinity effects on $T_m$ and $T_f$, the ice phenology modeled by FLake differs from the remote sensing observations. The discrepancy may be related to a number of factors not included in the model. One of them is the effect of salinity on the ice structure, density, and porosity; the others are precipitation, inflows, circulation under ice cover and wind, which is especially important for large-area lakes (Kirillin et al. 2012), such as Qinghai Lake. However, the air temperature apparently has the strongest effect on ice regime, especially in long-term changes, which appear to be well-simulated by FLake allowing us to study the effect of climate change on lake ice regime within the model ability." was added.

**More specific comments:**

**Comment #1**

**The "Study area" Section: The elevation of the Tibetan Plateau is never mentioned in the paper. What is the absolute elevation of Qinghai Lake surface? This is an important piece of information, please specify.**

**Author's response:** The elevation of the Qinghai Lake surface already given in brackets of the first sentence in "study area" section (Page 3 Line 17): "Qinghai Lake (36°32′–37°15′ N, 99°36′–100°47′ E, 3 194 m a.s.l.) is the biggest lake in China with a surface area of 4 497 km$^2$ (in 2017) and a catchment area of 29 660 km$^2$." Maybe it is not very obvious in brackets, so we have rewritten this part according to the Reviewer's suggestion.

**Relevant changes made in the manuscript:**

(1) Page 3 Line 17, ",3194 m a.s.l" was deleted.

(2) Page 3 Line19, the statements of "It is an endorheic, brackish lake (salinity 12.5 g l$^{-1}$, pH 9.3) (Deng et al., 2010) located on the northeast margin of the TP (Fig. 1)." Were corrected as "It is an endorheic, brackish lake (salinity 12.5 g l$^{-1}$, pH 9.3) (Deng et al., 2010) located on the northeast margin of the TP (Fig. 1) at the height of about 3 194 m a.s.l."

**Comment #2**

**P3 Lines 25-30: It follows from these numbers that the lake's water budget has been shifting towards an increase of the incoming components since 1970, accompanied by the decrease of evaporation. Then why the lake kept shrinking until 2004? Was the rate of shrinking in the 1960s much higher than in the early 2000s? Please explain.**

**Author's response:** We are very sorry for our unclear expression. The change in lake water level depends on the balance between incoming components and evaporation loss. Many studies on the water level changes in Qinghai Lake show that the water level in Qinghai Lake was declining in fluctuations from 1959 to 2004 (e.g. Zhang et al.,2011; Li et al., 2007; Tang et al., 2018; all cited in manuscript). This is because the water loss (i.e. evaporation) was generally larger than incoming water (e.g. runoff and precipitation) in this period, although the incoming components was increasing and the evaporation was decreasing (Tang et al., 2018), it still need some time to get balance with the water loss, so the lake kept shrinking until 2004 when the increasing incoming water balanced with the water loss. The rate of shrinking of Qinghai Lake in the early of 1960s indeed seems much higher than in the early 2000s (Zhang et al.,2011), but it turned to a temporary expanding in the late of 1960s (e.g. Zhang et al., 2011; Li et al., 2007). We have re-written this paragraph according to the Reviewer's suggestion.

**Relevant changes made in the manuscript:**

Page 3 Line 26, the statements of "Qinghai Lake is sensitive to climate change. The annual temperature of the Qinghai lake basin increased remarkably by about 0.3 °C per decade from 1961 to 2012 and the water level decreased at the average rate of 7.6 cm per year from 1961 to 2004 (Cui et al., 2016). According to the data from Gangcha station (the nearest meteorological station approximately 13 km north to Qinghai Lake), precipitation continuously increased in 1970-2015 by 15.603 mm per decade, especially after 2005; coupled with the melting of Qilian Mountain glaciers that increased the runoff to the lake, and the decreasing evaporation by 1.343 mm per year (Gangcha station) during 1970-2003 (Tang et al., 2018). Since 2004, the Qinghai lake stopped shrinking and began to expand, lake level increased at a rate of 14 cm per year during the period 2004-2012; and the regional climate gradually turned to the direction of "warm and humid" (Dong and Song, 2011; Zhang et al.,

2011, 2014b; Cui et al., 2016)." were corrected as "Qinghai Lake is sensitive to climate variability: Because the evaporation was generally larger than river runoff and precipitation from 1961 to 2004, the water level of Qinghai Lake decreased at an average rate of 7.6 cm per year (Cui et al., 2016). However, the precipitation continuously increased in 1970-2015 by 15.603 mm per decade according to the data from Gangcha station (the nearest meteorological station approximately 13 km north to

5      Qinghai Lake). Simultaneously, the runoff from the melting of Qilian Mountain glaciers was also increasing because of the regional warming trend of 0.319 °C per decade, coupled with the decreasing evaporation by 1.343 mm per year (observed by Gangcha station) during 1970-2003 (Tang et al., 2018). Since 2004, as the runoff and precipitation exceeded evaporation and the regional climate gradually turned to the direction of "warm and humid", the Qinghai lake level increased at a rate of 14 cm per year during 2004-2012 (Dong and Song, 2011; Zhang et al., 2011, 2014b; Cui et al., 2016)."

**Comment #3**

**P4 L15: "rare abnormal values influenced probably by cloud cover" – if you are confident that these abnormal values are artifacts corresponding to low clouds, then why keep them? Just remove them from your data base and the plot.**

**Author's response:** We have removed the abnormal values according to the referee's comment.

15    **Relevant changes made in the manuscript:**

(1) Page 4 Line 14, "We had removed few abnormal values that might be influenced by cloud cover (Langer et al., 2010)." was added.

(2) Page 4 Line 15, "except rare abnormal values influenced probably by cloud cover (Langer 15 et al., 2010)." was deleted.

20    **Comment #4**

**P5 Section 2.2: More details about the FLake model would be useful. What is the form of the expression for the profile in the lower layer?**

**Author's response:** We are very sorry for our negligence of the details about FLake. According to Reviewer's comment, we added the expression in FLake for the temperature profile in the lower layer.

25    **Relevant changes made in the manuscript:**

Page 5 Line 9, "The parameterization formula is:

$$\frac{\theta_s(t) - \theta(z,t)}{\Delta\theta(t)} = \Phi_\theta(\zeta) \quad h(t) \leq z \leq D \tag{1}$$

Where $t$ is time, $z$ is the depth, $\theta_s(t)$ is the temperature of the upper mixed layer of depth $h(t)$, $\Delta\theta(t) = \theta_s(t) - \theta_b(t)$ is the temperature differences across the thermally stratified layer of the depth of $\Delta h(t) = D - h(t)$, $D$ is the lake depth, $\theta_b(t)$ is the temperature at the lake bottom. $\Phi_\theta(\zeta)$ is a dimensionless "universal" function of the dimensionless depth $\zeta = \frac{z - h(t)}{\Delta h(t)}$ which

30    satisfies the boundary conditions $\Phi_\theta(0) = 0$ and $\Phi_\theta(1) = 1$. Based on the self-similarity assumption, the temperature profile can be expressed as a two-layer approximation:

$$\theta(t) = \begin{cases} \theta_s(t) & 0 \le z \le h(t) \\ \theta_s(t) - [\theta_s(t) - \theta_b(t))\Phi_\theta(\zeta)] & h(t) \le z \le D \end{cases} \qquad (2)$$

Substitution of Eq. (2) over the lake water column with subsequent substitution into the heat transport equation yields a set of ordinary differential equations, including lake in form of the shape factor $C_\theta = \int_0^1 \Phi_\theta(\zeta)$. The resulting equation system is complemented by an equation for evolution of the mixed layer depth $h(t)$, which is calculated based on the convective entrainment or relaxation-type equation in terms of wind mixing (see Mironov, 2008 for details).

The shape factor $C_\theta$ is parameterized by a relaxation formula:

$$\frac{dC_\theta}{dt} = sign\left(\frac{dh(t)}{dt}\right)\frac{C_\theta^{max} - C_\theta^{min}}{t_{rc}} \qquad C_\theta^{min} \le C_\theta \le C_\theta^{max} \qquad (3)$$

Where $t_{rc}$ is the empirically estimated relaxation time (s) of the temperature profile in the thermocline from one limiting curve to the other, following the change of sign in $\frac{dh(t)}{dt}$. $C_\theta^{min} = 0.5$ and $C_\theta^{max} = 0.8$ are the minimum and maximum values of the shape factor." was added.

**Comment #5**

**P5 L30 and thereafter: The adjustments introduced to the air temperature and wind speed through linear regressions seem to help very little in minimizing biases between the simulated and the observed LST, so what is the point of using them?**

**Author's response:** We are very sorry for our unclear expression. The adjustments allowed reducing the bias and rms by half in summer and autumn (see Section 3.1 of the revised manuscript). However, the buoy observation data available for bias correction were unfortunately not complete, covering only summer and autumn and some of the data are missing. No correction was performed for other parts of the year. Therefore, an appreciable bias remained in the results. In addition, the revised air temperature and wind speed may not have enough consistency to use, but it helps in understanding and evaluating the bias caused by the forcing data. We have made correction according to the Reviewer's comments.

**Relevant changes made in the manuscript:**

Page 6 Line 2, "Through the correction of the driving data, we found that positive bias between simulated LST and satellite data can be partly explained by the differences in the forcing weather data measured over the lake and provided by the ITPCAS data." was added.

**Comment #6**

**P11 L15: "Keeping in mind the cool skin effect, we can suggest that the model predictions of the bulk LST are even better than the satellite data suggest" – But your Figure 2 shows good agreement between the satellite and the buoy data, and the latter measured bulk temperature. Therefore, it looks like the skin effect in this case did not affect much**

**the satellite-derived temperatures.**

**Author's response:** We are very sorry for our incorrect writing. The satellite observed LST have a little negative bias (-0.36 °C, in summer and autumn) compared with the buoy bulk temperature. While the y-scale of Fig. 2 do not allow to see the bias clearly, the bias values are added to the panels on the figure. If the skin effect, in this case, did not affect much the satellite-derived temperatures, it probably caused by the different measurement methods between satellite and buoy.

**Relevant changes made in the manuscript:**

(1) Page 11 Line 13, the statements of "This discrepancy was apparently contributed by the cool skin effect (Crosman and Horel, 2009)" were corrected as "This discrepancy may partly be contributed by the cool skin effect (Crosman and Horel, 2009)"

(2) Page 11 Line 15, the statements of "Keeping in mind the cool skin effect, we can suggest that the model predictions of the bulk LST are even better than the satellite data suggest, though exact estimation of the cool skin correction is out of the scope of this study." were corrected as "This suggest that the model predictions of the bulk LST may be better than comparison against the satellite data shows, though exact estimation and correction of the cool skin effect is out of the scope of this study."

**Responses to the comments from the Referee #2**

**The Qinghai lake is the largest in land lake in China. It has large volume of biotic resources and tourism resources. Its thermodynamic changes under global warming remains unclear. Su et al. use a one-dimensional lake model to investigate thermodynamic changes of the Qinghai lake in the last three decades. The results show that the Qinghai lake has been warming up in the last three decades and the warming was the strongest in winter. Before getting published, however, this manuscript should be revised in several aspects. Please consider the points listed below and marked out in the manuscript. I strongly recommend language editing by some native English speaker, there are many errors in the grammar and improper expressions.**

**Comment #1**

**The authors emphasize the ice cover plays the first role in long-term change of thermodynamics, however, they do not validate the performance of Flake on the ice dynamics. The ice-on and ice-off dates can be obtained from MODIS data. The authors can use the MODIS-derived ice-on and -off dates to validate the performance of the Flake on ice phenology. Here is the data link: http://www.csdata.org/p/214/.**

**Author's response:** Thanks for the referee's suggestion. As referee's suggestion, we use the MODIS-derived ice-on and ice-off dates to validate the FLake on ice phenology.

**Relevant changes made in the manuscript:**

(1) Page 4 Line 20, section "2.2.3 Dataset of lake ice phenology in Qinghai Lake" was added, and the section "2.2.3 ITPCAS Forcing Data" was changed to "2.2.4 ITPCAS Forcing Data"

(2) Page 4 Line 21, the content of section "2.2.3 Dataset of lake ice phenology in Qinghai Lake" in revised manuscript as "The dataset on lake ice phenology in Qinghai Lake from 2000 to 2018 was built by using RS and GIS technologies based on Terra MODIS surface reflectance product and Landsat TM/ETM+/OLI remote sensing images (Qi et al., 2018). The dataset uses the method of threshold segmentation to extract the ice area of Qinghai Lake based on MOD09GQ product by setting a reflectance threshold for the red band and a reflectance difference threshold between red and near-infrared bands. The extracted ice area was then validated against the visually interpreted ice area based on Landsat TM/ETM+/OLI images. The dataset includes ice-water vector boundary data, area ratio, and phenological characters in Qinghai Lake from 2000 to 2018. Phenological information includes the start and end dates of lake freeze-up and break-up, and ice cover duration. The dataset provides a reference for exploring the spatio-temporal characteristics of lake ice in Qinghai Lake, as well as for estimating lake ice cover response to climate changes in the region." was added.

(3) Page 26 Figure 7, in the revised manuscript, the MODIS-derived ice phenology was added.

[Figure]

**Figure 7: The interannual variations of simulated annual maximum ice thickness (a), freeze-up date (b), break-up date (c) and ice duration (d) of Qinghai Lake. The coloured (grey) line indicate SL (FL) experiment. The red dash line is air temperature anomaly in the specified period and the black line is ice phenology observation derived from the satellite.**

5    (4) Page 7 Line 6 in section 3.3 Lake ice cover, "Compared with FL experiment, the salinity parameterization for Tm and Tf in SL experiment has a certain effect on the ice phenology (Fig. 7): the maximum ice thickness is reduced, the freeze-up date is delayed and the break-up date is advanced, leading to a shorter ice duration period. Nevertheless, the interannual changes between them remained consistent. The simulated freeze-up and break-up date in FL and SL experiments are both later than satellite observations, with some differences in interannual variations but similar range in ice duration. In SL experiment, the

10    maximum ice thickness and the break-up date are closer to the observations, the former was reported of 0.7 m by Chen et al (1995). Hence the ice phenology results from SL experiment were used for further analysis." was added.

Page 10 Line 18, in section 4.1 Model performance, "Despite incorporation of the salinity effects on Tm and Tf improved simulation accuracy of maximum ice thickness and break-up date, the ice phenology modeled by FLake still differs from the remote sensing observations. The discrepancy may be related to a number of factors not included in the model. One of them

15    is the effect of salinity on the ice structure, density, and porosity; the others are precipitation, inflows, circulation under ice cover and wind, which is especially important for large-area lakes (Kirillin et al. 2012), such as Qinghai Lake. However, the air temperature apparently has the strongest effect on ice regime, especially in long-term changes, which appear to be well-simulated by FLake allowing us to study the effect of climate change on lake ice regime within the model ability." was added.

20    **Comment #2**

**The author should deemphasize the purpose to validate the performance of FLake on the Tibetan Plateau. Because both Lazhu et al. (2016) and Kirillin et al. (2017) has demonstrated its performance on the Tibetan Plateau. In Lazhu's study, the Nam Co lake is also a brackish and large lake. They even use observational temperature at different depths**

**to validate its performance. In this respect, their study should be a better case to evaluate the performance of the Flake Model.**

**Author's response:** Thanks for the Referee's kind advice. We agree with the reviewer: the studies of Lazhu et al (2016) and Kirillin et al (2017) considered several aspects of the performance of FLake on the Tibetan Plateau lakes. The first study was focused on the evaporation estimations at Nam Co Lake (salinity ~1.78 g l$^{-1}$), the second one considered thermal regime of freshwater lakes. The present study comprehensively tests the FLake performance on the largest, brackish lake of the Plateau as a "worst-case" test for a 1-D freshwater model. It suggests that the results are extendable on the vast majority of the Tibetan lake system with at least the same or better performance. And in the revised manuscript, the salinity effects on temperature of maximum density and freezing point had been considered based on the comments, and the model performance has been improved. According to the reviewer's suggestion, we deemphasized the purpose to evaluate Flake and added the research of Lazhu et al. (2016) and Kirillin et al. (2017), but still mentioned one of our study purposes was to evaluate the FLake model in the Tibetan Plateau.

**Relevant changes made in the manuscript:**

(1) Page 3 Line 3, in section 1 Introduction, the statements of "FLake was numerously tested before for different lakes worldwide (Kirillin, 2010; Bernhardt et al., 2012; Stepanenko et al., 2013; Thiery et al., 2014), including freshwater lakes of the TP (Kirillin et al., 2017)." were corrected as "The model was numerously tested before for different lakes worldwide (Kirillin, 2010; Bernhardt et al., 2012; Stepanenko et al., 2013; Thiery et al., 2014), including freshwater lakes (Kirillin et al., 2017) and a brackish lake (Lazhu et al, 2016) on the TP."

(2) Page 10 Line 18, in section 4.1 Model performance, the statements of "The good prediction of the LST over the Tibet by the relatively simple, highly parameterized model FLake, verified by satellite and buoy data, is one of the core results of this study." were corrected as: "Following the studies of Lazhu et al., (2016) on Nam Co Lake (salinity ~1.78 g l$^{-1}$) and Kirillin et al. (2017) on freshwater Lakes Ngoring and Gyaring, the good prediction of the LST over the largest, brackish lake of TP by the relatively simple, highly parameterized model FLake, verified by satellite and buoy data, is one of the core results of this study."

**Comment #3**

**Even the Qinghai lake is a brackish lake, but its salinity is not low (~12.5 g/L). I agree with the authors that the salinity would not change the mixing type (dimictic), but salinity produces effects on the dates of spring and autumn overturning, which will change the energy flux. do the authors have any other observation data related the mixing of lake water column? If they have, they should show it or have some description on it.**

**Author's response:** Thanks for the referee's good suggestion. Considering the referee's suggestion, we gave rerun the FLake model extended by simple parameterizations of salinity effect on temperature of maximum density and freezing point (see the reply to the first comment of Referee #1), after considering the salinity effect, the dates of spring and autumn overturning changed. The energy fluxes were also reanalyzed based on the new simulation results, but the main conclusions did not change.

More observation data on mixing of lake water column are required for a more thorough analysis of salinity influence on stratification, but we do not have any other observation data related to the mixing of lake water column so far.

**Relevant changes made in the manuscript:**

Page 10 Line 16, in Section 4.1 Model performance, as described above in response to Referee #1, the statements "Still, since the model used here is the freshwater lake model, the simulated mixing regime may have some difference with the actual situation of Qinghai Lake." were corrected as "Salinity can influence the temperature of maximum density ($T_m$) and the freezing temperature of water ($T_f$). According to the 12.5g $l^{-1}$ salinity of Qinghai Lake, these two parameters equal to 1.28 °C and -0.69 °C instead of the default model configurations of 4 °C and 0 °C respectively. Considerations of the salinity effects lead to a slightly earlier spring overturn and a later autumn overturn, and consequently to an extension of the lake stratification period. Because the salinity stratification effects cannot be completely included in the model designed for freshwater lakes, the simulated mixing regime may have some differences from the actual situation of Qinghai Lake."

**I have some other comments and suggestions, please find the attached PDF file for details.**

**Comments in the supplement:**

**Comment #1**

**P3 L20: "The lake usually freezes up in December/January and the ice breaks up in early April." Where dose these data come from? Your observation or reference.**

**Author's response:** We are very sorry for our unclear expression. These data are come from the published paper by Li et al (2016) and already included in this paragraph.

**Relevant changes made in the manuscript:**

Page 3 Line 20, the statements of "The lake usually freezes up in December/January and the ice breaks up in early April. The average annual lake water temperature is 5.4 °C, with a maximum monthly temperature of 17.2 °C (August) and a minimum of -2.0 °C (January) (Li et al., 2016)." were corrected as "The lake is ice-covered from December/January to early April; the average annual lake water temperature is 5.4 °C, with the maximum monthly temperature of 17.2 °C (August) and the minimum of -2.0 °C (January) (Li et al., 2016)."

**Comment #2**

**P4 L5: Do you have water temperature observations on different depths? If you have, you'd better show it.**

**Author's response:** No, we don't have the observations on different depths so far.

**Comment #3**

**P5 L5: What is the version of the Flake model?**

**Author's response:** We download the official version of FLake from the model's website http://www.flake.igbberlin.de/index.shtml, and have already described it in section Data availability.

**Comment #4**

**P5 L12: You do not have to emphasize the computational efficiency to much which has been emphasized in row-5, page-3. Considering the computing ability of current PCs, almost all one-dimensional or two-dimensional models can be well executed.**

**Author's response:** It is really true as Reviewer suggested that we do not have to emphasize the computational efficiency to much which has been emphasized in row-5, page-3. We have made correction according to the Reviewer's comments.

**Relevant changes made in the manuscript:**

Page 5 Line 12, "This simple two-layer parameterization of the water column provides FLake with computational efficiency, while preserves the essential physics." was deleted.

**Comment #5**

**P7 L5: what does this mean? do you mean the break-up and freeze-up dates are the most sensitive proxy to all the meteorological parameters, including air temperature, solar radiation, and wind?**

**Author's response:** We are very sorry for our incorrect writing. We have made correction according to the Reviewer's comments.

**Relevant changes made in the manuscript:**

Page 7 Line 5, the statements of "The variations of break-up and freeze-up dates are sensitive to the meteorological conditions, in the first place, air temperature, solar radiation, and wind (Duguay et al.,2006; Latifovic et al., 2007; Ye et al., 2011; Kirillin et al., 2012; Yao et al., 2016)." were corrected as "The variations of break-up and freeze-up dates are sensitive to the meteorological conditions, e.g. air temperature, solar radiation, and wind (Duguay et al.,2006; Latifovic et al., 2007; Ye et al., 2011; Kirillin et al., 2012; Yao et al., 2016)."

**Comment #6**

**P7 L16: What is the direction of the energy flux? Outgoing-negative, incoming-positive?**

**Author's response:** For radiation flux and net energy flux, downward is positive and upward is negative. For SH, LH and lake surface released heat flux, upward is positive and downward is negative.

**Comment #7**

**P8 L2: I am confused by the logic of this part. based on the first sentence, I thought that the authors wanted to discuss how the lake-air temperature difference determined the energy flux. but after reading the following two paragraphs, I realized that the authors discussed how the different fractions of the energy flux contributed to the lake-air temperature difference.**

**Author's response:** We are very sorry for our incorrect writing. We have made correction according to the Reviewer's comments.

**Relevant changes made in the manuscript:**

Page 8 Line 2, "The lake-air temperature difference is one of the important factors determining the surface heat exchange." was deleted.

**Comment #8**

**P11 L20: I was confused by this paragraph. as the Figure 6 shows, not only air temperature and wind speed, but also long-wave radiation and short-wave radiation have obvious correlation with water temperature. Long-wave radiations even have the highest correlation coefficient. Why the author only claims changes in windspeed and air temperature drive the lake warming? Furthermore, the net long-wave radiation has a trend of -1.86 W/m$^2$/decade, the net short-wave radiation has a trend of 2.35 W/m$^2$/decade based on the Figure 8. These trends are much larger the trends of sensible heat flux and latent heat flux which are functions of lake-air temperature difference and wind speed. Please consider the roles of short- and long-radiation in lake warming, and add corresponding stuff in your conclusion.**

**Author's response:** The air temperature and longwave radiation are obviously correlated, because the downward longwave radiation is function of the air temperature in the fourth degree (affected by emissivity properties of the atmosphere). The annual shortwave radiation is negatively and insignificantly (R<0.3) correlated with the water temperature, also can explain the slower increase of the water temperature compared with the air temperature (see Kirillin et al. 2017). Hence, air temperature remains to be the major factor affecting the long-term trend in water temperatures. We have made correction according to the Reviewer's comments.

**Relevant changes made in the manuscript:**

(1) Page 11 Line 20, the statements of "As expected, the correlation analysis shows that the changes in LST are closely related to air temperatures, downward longwave radiation and wind speed (Fig. 6). The increase of the air temperature and downward long-wave radiation plays a key role in lake surface temperature warming, and the decrease in wind speed also promoted the warming of the lake surface temperature." were revised as "As expected, the correlation analysis shows that the changes in LST are closely related to air temperatures, downward longwave radiation and wind speed (Fig. 6). The increase of the air temperature and downward long-wave radiation plays a key role in lake surface temperature warming, and the decrease in wind speed also promoted the warming of the lake surface temperature."

(2) Page 11 Line 21, "The downward shortwave radiation is negatively and insignificantly (R<0.3) correlated with the water temperature that also can explain the slower increase of the water temperature compared with the air temperature (see Kirillin et al. 2017). The decrease in ice cover duration increases in turn the annual amount of shortwave radiation penetrating to the water column, accelerating the net warming." was added.

**Comment #9**

**P12 L20: For low-altitude lakes in the mid-latitudes? you'd better list at least one reference paper here.**

**Author's response:** We have re-written this part according to the Reviewer's suggestion.

**Relevant changes made in the manuscript:**

Page 12 Line 20, the statements of "For low-altitude lakes in other regions, the air temperature is typically higher than LST between the ice melt and the temperature equilibration in the mid-summer. In the later surface cooling in autumn, the LST remain higher than the air temperature down to the ice on. The seasonal pattern remains valid for both large (Rouse et al., 2003) and small (Nordbo et al., 2011) lakes." were corrected as "For low-altitude temperate and boreal lakes, the air temperatures are typically higher than LST after the ice-off and remain higher until temperature equilibrates around mid-summer. In the subsequent period down to ice-on, the LSTs are typically higher than the air temperatures. Hence, the atmospheric boundary layer is generally stable throughout much of the summer season over low-altitude lakes (Scott and Huff, 1996; Rouse et al., 2003; Gianniou and Antonopoulos, 2007; Momii and Ito, 2008; Nordbo et al., 2011)."

**Comment #10**

**P26 Fig. 7: Do you have observational data of ice thickness? ~1.1 m is very thick for lake ice. The thickest ice on the Namco lake is 70 cm based on the observation (Qu et al., 2012, Chinese with English abstract, Lake Ice and Its Effect Factors in the Nam Co Basin, Tibetan Plateau) where is 1500 m higher than the Qinghai lake.**

**Author's response:** We do not have observational data of ice thickness, but according to the study of Chen et al (1995), the maximum ice thickness in Qinghai Lake is about 0.7 m. The simulated maximum ice thickness of ~1.1 m was the situation in 1980s, in the later 2000s, the thickness of the lake ice had dropped significantly to about 0.7 m, same with Namco in this period according to Qu et al (2012). And after considering the salinity effect, the simulated maximum ice thickness further reduced (Fig. 7 in revised manuscript). Moreover, many factors affecting lake regime have not been considered by FLake. We have made some explanations according to the Reviewer's comments.

**Relevant changes made in the manuscript:**

(1) Page 7 Line 5, in section 3.3 Lake ice cover, "In SL experiment, the maximum ice thickness and the break-up date are closer to the observations, the former was reported of 0.7 m by Chen et al (1995)." was added.

(2) Page 10 Line 18, in section 4.1 Model performance, "Despite incorporation of the salinity effects on $T_m$ and $T_f$ improved simulation accuracy of maximum ice thickness and break-up date, the ice phenology modeled by FLake still differs from the remote sensing observations. The discrepancy may be related to a number of factors not included in the model. One of them is the effect of salinity on the ice structure, density, and porosity; the others are precipitation, inflows, circulation under ice cover and wind, which is especially important for large-area lakes (Kirillin et al. 2012), such as Qinghai Lake." was added.

**Comment #11**

**P28: Do not understand what the blue bars and red bars represent respectively in Figure 9a**

**Author's response:** We are very sorry for our negligence and have made correction according to the Reviewer's comments.

**Relevant changes made in the manuscript:**

Page 28 Figure 9, the statements of "Figure 9: Climatological mean seasonal variations (5-day moving average, lines) in simulated LST and air temperature (a) with their difference (b), downward shortwave radiation(c), downward longwave radiation(d), net shortwave radiation (e) and net longwave radiation (f). The bars indicate monthly averaged mean annual variation trend (red for positive and blue for negative) from 1979 to 2012. Solid points at end of the bars mean pass significance test of $p<0.01$ and hollow points mean $p<0.05$. The grey areas indicate the freeze-up and break-up date variation range of the lake." were corrected as "Figure 9: Climatological mean seasonal variations (5-day moving average, lines) in simulated LST and air temperature (a) with their difference (b), downward shortwave radiation (c), downward longwave radiation(d), net shortwave radiation (e) and net longwave radiation (f) at lake surface. The bars indicate their monthly averaged mean annual variation trend from 1979 to 2012, red for positive and blue for negative except in (a) that for air temperature and LST respectively. Solid points at end of the bars mean pass the significance test of $p<0.01$ and hollow points mean $p<0.05$. The grey areas indicate the freeze-up and break-up date variation range of the lake."

**Other changes:**

We tried our best to improve the manuscript and made some changes in the manuscript. These changes will not influence the content and framework of the paper. Many other changes not list here can be found in the marked-up manuscript version enclosed below.

We appreciate for Editors/Reviewers' warm work earnestly, and hope that the correction will meet with approval.

Once again, thank you very much for your comments and suggestions.

[revised manuscript text omitted]